# Aggregated Sharpness-Aware Minimization Is Suboptimal in Domain Generalization: Towards Per-Domain Sharpness-Aware Minimization

## Abstract

Domain generalization (DG) aims to learn models that perform well on unseen target domains by training on multiple source domains. Sharpness-Aware Minimization (SAM), known for finding flat minima that improve generalization, has therefore been widely adopted in DG. However, we argue that the prevailing approach of applying SAM to the aggregated loss for domain generalization is fundamentally suboptimal. This "aggregated sharpness" objective can be deceptive, leading to convergence to fake flat minima where the total loss surface is flat, but the underlying per-domain landscapes remain sharp. To establish a more principled objective, we analyze a worst-case risk formulation that reflects the true nature of DG. Our analysis reveals that per-domain sharpness provides a valid upper bound on this risk, while aggregated sharpness does not, making it a more theoretically grounded target for robust domain generalization. Motivated by this, we propose *Domain-wise Gradual SAM (DGSAM)*, which applies gradual, domain-wise perturbations to effectively control per-domain sharpness in a computationally efficient manner. Extensive experiments demonstrate that DGSAM not only improves average accuracy but also reduces performance variance across domains, while incurring less computational overhead than SAM.

## 1 Introduction

Deep neural networks achieve remarkable performance under the independent and identically distributed (i.i.d.) assumption (Kawaguchi et al., 2017), yet this assumption often fails in practice due to *domain shifts*. For example, in medical imaging, test data may differ in acquisition protocols or device vendors (Li et al., 2020), and in autonomous driving, variations in weather or camera settings introduce further domain shifts (Khosravian et al., 2021). Since it is impractical to include every possible scenario in the training data, *domain generalization* (DG) seeks to learn models that generalize to unseen target domains using only source domain data (Muandet et al., 2013; Arjovsky et al., 2019; Li et al., 2018c; Volpi et al., 2018; Li et al., 2019).

A common DG strategy is to learn domain-invariant representations by aligning source domain distributions and minimizing their discrepancies (Muandet et al., 2013; Arjovsky et al., 2019), adversarial training (Li et al., 2018c; Ganin et al., 2016), data augmentation (Volpi et al., 2018; Zhou et al., 2020; 2021), and meta-learning approaches (Li et al., 2019; Balaji et al., 2018). These strategies share the common goal of solving the core challenge of DG: learning from source domains with structured shifts (e.g., artistic style, weather conditions) to generalize to unseen variations of these structures. More recently, flat minima in the loss landscape have been linked to improved robustness under distributional shifts (Cha et al., 2021; Zhang et al., 2022; Chaudhari et al., 2019). In particular, Sharpness-Aware Minimization (SAM) (Foret et al., 2021) perturbs model parameters along high-curvature directions to locate flatter regions of the loss surface, and has been applied to DG (Wang et al., 2023; Shin et al., 2024; Zhang et al., 2024).

However, we argue that the prevailing approach of applying SAM to the aggregated loss is fundamentally suboptimal. Our analysis reveals that the current SAM-based approach for DG pursues an

unrealistic goal: robustness to perturbations of a probabilistic average of the source domains, rather than the coherent shifts of per-domain source types that characterize real-world DG. This misalignment can be deceptive, leading to convergence to *fake flat minima* that appear flat on aggregated loss but remain sharp on separate domains. We find this occurs because aggregated sharpness is an unreliable proxy for the per-domain flatness that is truly required for robust generalization. To establish a more principled objective, we introduce a worst-case risk formulation that formalizes this notion of coherent shifts. We then theoretically demonstrate that per-domain sharpness, not aggregated sharpness, provides a valid upper bound on this risk, making it a more grounded target for optimization.

Motivated by these insights, we propose a novel DG algorithm, **Domain-wise Gradual Sharpness-Aware Minimization (DGSAM)** that employs a gradual and domain-specific perturbation mechanism designed to effective control per-domain sharpness. DGSAM improves upon existing SAM-based DG methods in three key aspects. First, it efficiently reduces the per-domain sharpness of source domains rather than the aggregated sharpness of the total loss, enabling better learning of domain-invariant features. Second, it achieves high computational efficiency by reusing gradients computed during gradual perturbation, in contrast to traditional SAM-based methods that incur twice the overhead of standard empirical risk minimization. Third, while prior approaches rely on proxy curvature metrics, DGSAM controls the eigenvalues of the Hessian, which are the most direct indicators of sharpness (Keskar et al., 2016; Ghorbani et al., 2019b). Our extensive experiments confirm the superiority of this approach. DGSAM demonstrates a superior balance of accuracy and robustness, achieving the highest average accuracy and the lowest average domain-wise variance across five benchmarks. Furthermore, DGSAM shows broad compatibility by enhancing various DG frameworks and confirms its scalability on large-scale Vision Transformer models, all while being more computationally efficient than standard SAM.

## 2 PRELIMINARIES AND RELATED WORKS

### 2.1 DOMAIN GENERALIZATION

Let $\mathcal{D}_s := \{\mathcal{D}_i\}_{i=1}^{S}$ denote the collection of training samples, where $\mathcal{D}_i$ represents the training samples from the $i$-th domain[1]. The total loss over all source domains is defined as:

$$\mathcal{L}_s(\theta) := \frac{1}{|\mathcal{D}_s|} \sum_{\mathcal{D}_i \in \mathcal{D}_s} \mathcal{L}_i(\theta), \tag{1}$$

where $\mathcal{L}_i$ denotes the loss evaluated on samples from the $i$-th domain, and $\theta$ is the model parameter.

A naïve approach to DG minimizes the empirical risk over the source domains.: $\theta_s^* = \arg\min_\theta \mathcal{L}_s(\theta)$. However, this solution may fail to generalize to unseen target domains, as it is optimized solely on the training distribution. The goal of domain generalization is to learn parameters $\theta$ that are robust to domain shifts, performing well on previously unseen domains.

As the importance of DG has grown, several datasets (Li et al., 2017b; Fang et al., 2013; Peng et al., 2019) and standardized protocols (Gulrajani & Lopez-Paz, 2021; Koh et al., 2021) have been introduced. Research directions in DG include domain-adversarial learning (Jia et al., 2020; Li et al., 2018c; Akuzawa et al., 2020; Shao et al., 2019; Zhao et al., 2020), moment-based alignment (Ghifary et al., 2016; Muandet et al., 2013; Li et al., 2018b), and contrastive loss-based domain alignment (Yoon et al., 2019; Motiian et al., 2017). Other approaches focus on data augmentation (Xu et al., 2020; Shi et al., 2020; Qiao et al., 2020), domain disentanglement (Li et al., 2017a; Khosla et al., 2012), meta-learning (Li et al., 2018a; Zhang et al., 2021; Li et al., 2019), and ensemble learning (Cha et al., 2021; Seo et al., 2020; Xu et al., 2014).

### 2.2 SHARPNESS-AWARE MINIMIZATION

A growing body of work connects generalization to the geometry of the loss surface, especially its curvature (Hochreiter & Schmidhuber, 1994; Neyshabur et al., 2017; Keskar et al., 2017; Chaudhari et al., 2019; Foret et al., 2021). Building on this, Foret et al. (2021) proposed Sharpness-Aware

---

[1]With slight abuse of notation, we also use $\mathcal{D}_i$ to represent the underlying data distribution of the $i$-th domain.

Minimization (SAM), which optimizes the model to minimize both the loss and the sharpness of the solution. The SAM objective is defined as:

$$\min_{\theta} \max_{\|\epsilon\| \leq \rho} \mathcal{L}(\theta + \epsilon), \tag{2}$$

where the inner maximization finds the worst-case perturbation $\epsilon$ within a neighborhood of radius $\rho$.

Following the success of SAM, several extensions have emerged, primarily focusing on refining the sharpness surrogate (Kwon et al., 2021; Zhuang et al., 2022; Zhang et al., 2022) or reducing its computational overhead (Du et al., 2022; Liu et al., 2022; Mordido et al., 2024). The promise of improved generalization has naturally led to the exploration of sharpness-aware methods in domain generalization. A common strategy is to apply SAM to the aggregated loss over source domains (Wang et al., 2023; Shin et al., 2024; Cha et al., 2021; Dong et al., 2024), which seeks a solution that is flat with respect to the total aggregated loss. Recognizing the importance of domain-level structure, recent work has incorporated domain information, either by adding regularization to penalize inter-domain loss variance (Zhang et al., 2024) or by iteratively refining loss landscapes for consistency across domains (Li et al., 2025).

While these approaches represent important progress, they either still optimize for aggregated sharpness or implicitly encourage per-domain flatness through consistency constraints without a formal per-domain sharpness minimization objective. In the following section, we propose a domain-wise objective that explicitly minimizes the sharpness within each domain's loss landscape. A more detailed categorization and comparison of existing approaches is provided in Appendix H.

## 3 RETHINKING SHARPNESS IN DOMAIN GENERALIZATION

The prevailing paradigm in the current literature is to apply SAM to the aggregated loss across all source domains. We argue this approach is fundamentally suboptimal for domain generalization, as it is built on an assumption that is misaligned with the core nature of the DG problem itself. By collapsing the crucial structural information between domains, this strategy shifts the optimization objective from learning features that are truly domain-invariant, to merely seeking robustness for a probabilistic average of the source domains. This is a critical distinction, as this probabilistic average may not represent any realistic domain and is not equivalent to the shared, invariant features required for true generalization. This misalignment can be deceptive, leading to convergence to 'fake flat minima'. In Section 3.1, we first provide a formal and intuitive illustration of this pitfall. We then propose a more principled objective grounded in a worst-case risk formulation that respects this essential domain-specific structure in Section 3.2.

### 3.1 AGGREGATED SHARPNESS PITFALLS: THE FAKE FLAT MINIMA PROBLEM

To formalize our perspective, we distinguish between two key concepts. The prevailing approach for SAM in DG focuses on aggregated sharpness, defined as:

$$\mathcal{S}_{\text{agg}}(\theta; \rho) = \max_{\|\epsilon\| \leq \rho} \big( \mathcal{L}_{\text{s}}(\theta + \epsilon) - \mathcal{L}_{\text{s}}(\theta) \big).$$

where $\mathcal{L}_{\text{s}}$ is the total loss over all source domains, defined in equation 2.1. In contrast, our work focuses on the per-domain sharpness of each source domain $\mathcal{D}_i$, defined as:

$$\mathcal{S}_i(\theta; \rho) = \max_{\|\epsilon\| \leq \rho} \big( \mathcal{L}_i(\theta + \epsilon) - \mathcal{L}_i(\theta) \big).$$

To generalize well to unseen domains, a model must learn representations that are robust to various domain shifts. The most direct way to achieving this is to ensure that the learned solution is robust against new domains that are variations of each of the source domains seen during training. Therefore, an ideal DG approach should find a solution that is simultaneously flat with respect to every source domain, a property directly captured by per-domain sharpness ($\mathcal{S}_i$).

The prevailing approach of minimizing aggregated sharpness ($\mathcal{S}_{\text{agg}}$), however, does not guarantee this ideal outcome. As aggregated sharpness is measured on the aggregated loss, it is possible for this mixture to be flat while the loss landscapes of the underlying separate domains remain sharp. This presents a critical failure mode: if an unseen test domain shares characteristics with a source domain

for which the model has high per-domain sharpness, the model will likely fail, regardless of its low aggregated sharpness. This divergence, where low aggregated sharpness masks high per-domain sharpness, leads to what we term *fake flat minima*. The following proposition formally demonstrates that aggregated and per-domain sharpness are not necessarily correlated.

**Proposition 3.1.** *Let $\theta$ be a model parameter and $\rho > 0$ a fixed perturbation radius. Then, there exist two local minima $\theta_1$ and $\theta_2$ such that*

$$\mathcal{S}_{agg}(\theta_1; \rho) < \mathcal{S}_{agg}(\theta_2; \rho) \quad but \quad \frac{1}{S}\sum_{i=1}^{S}\mathcal{S}_i(\theta_1; \rho) \geq \frac{1}{S}\sum_{i=1}^{S}\mathcal{S}_i(\theta_2; \rho).$$

*Equivalently,*

$$\mathcal{S}_{agg}(\theta_1; \rho) < \mathcal{S}_{agg}(\theta_2; \rho) \implies \frac{1}{S}\sum_{i=1}^{S}\mathcal{S}_i(\theta_1; \rho) < \frac{1}{S}\sum_{i=1}^{S}\mathcal{S}_i(\theta_2; \rho).$$

The proof is deferred to Appendix B.1. This proposition provides the formal basis for the fake flat minima phenomenon, confirming that a low value of aggregated sharpness ($\mathcal{S}_{\text{agg}}$) can be achieved even when the average per-domain sharpness $\left(\frac{1}{S}\sum_i \mathcal{S}_i\right)$ remains high.

To illustrate this phenomenon, we present a 2-dimensional toy example involving two domains and two loss functions. Each domain shares the same base loss shape (Figure 2a) but is shifted along one axis. Figures 2b and 2c visualize the total loss from two perspectives. In this example, region **R1** corresponds to an *ideal solution*, where both single domain losses exhibit flat minima. In contrast, region **R2** remains sharp for each single domain loss, but appears deceptively flat in the total loss due to cancellation of opposing sharp valleys (Figure 1). As a result, both SAM and SGD converge to region **R2** (Figure 2d), which constitutes a *fake flat minimum*.

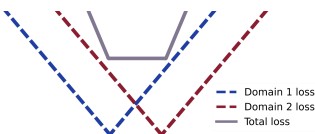

Figure 1: Fake flat minimum: two sharp per-domain losses (dotted) cancel out when summed, resulting in a deceptively flat total loss (solid).

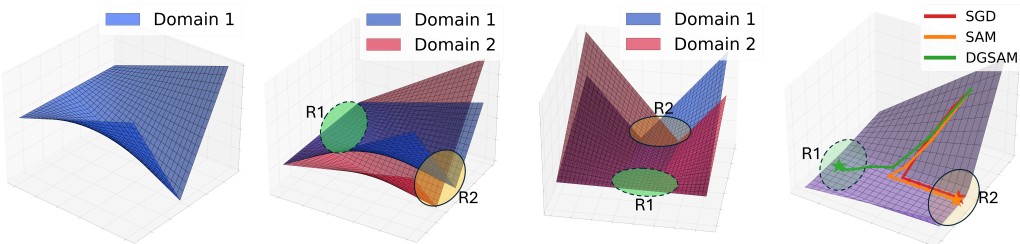

(a) Loss landscape of a single domain

(b) Side view of the total loss landscape

(c) Rear view of the total loss landscape

(d) Optimization trajectories

Figure 2: Toy example: two conflicting loss functions construct two different type of flat minima. An interactive visualization of toy example is available at `https://dgsam-toy-example.netlify.app/`.

The pitfall of the aggregated sharpness approach is not merely an theoretical concept. We confirm this phenomenon in practical DG tasks using ResNet-50 on the PACS dataset. As visualized in Appendix E, while SAM produces minima that are flat with respect to the total loss, the loss landscapes for the separate domains remain sharp, providing direct empirical evidence of the fake flat minima problem.

## 3.2 PER-DOMAIN SHARPNESS: A PRINCIPLED OBJECTIVE FOR DG

To establish a principled objective for SAM in DG, we need to define a performance measure that truly reflects the challenges of the task. As we have argued, a true domain shift is not a random

perturbation of the averaged sources. For instance, a model trained on 'Photo' and 'Sketch' domains is not evaluated on their pooled mixture, but rather on a new, coherent domain such as 'Cartoon' or 'Watercolor painting'. This new domain represents a coherent shift from one of the existing styles, not a deviation from their probabilistic mixture. A truly robust model, therefore, must be resilient to the worst-case shift originating from any of the each source domains it was trained on.

Based on this principled view, we now formalize the average worst-case domain risk. Let $\{\mathcal{D}_i\}_{i=1}^S$ denote the source distributions. For each source domain $i$, we define the local uncertainty set of potential target domains as:

$$\mathcal{U}_i^\delta = \left\{ \mathcal{D} : \mathrm{Div}(\mathcal{D} \| \mathcal{D}_i) \leq \delta \right\},$$

where $\mathrm{Div}(\cdot \| \cdot)$ is a divergence measure (e.g., KL-divergence, Wasserstein distance). This set $\mathcal{U}_i^\delta$ contains all unseen target domains that lie within a divergence $\delta$ of the source domain $\mathcal{D}_i$. The average worst-case domain risk is then the expected risk under the worst-case shift from each source domain:

$$\mathcal{E}(\theta; \delta) := \frac{1}{S} \sum_{i=1}^S \sup_{\mathcal{D} \in \mathcal{U}_i^\delta} \mathcal{L}_\mathcal{D}(\theta).$$

This principled risk formulation allows us to formally investigate which notion of sharpness, aggregated or per-domain, serves as a better optimization target.

**Theorem 3.2.** *Let $\mathcal{L}_s(\theta)$ denote the total loss over all source domains, $\mathcal{S}_{agg}(\theta; \rho)$ the aggregated sharpness, and $\mathcal{S}_i(\theta; \rho)$ the per-domain sharpness for the $i$-th domain. Then, for all $\theta$ and $\rho \geq \rho(\delta)$,*

$$\mathcal{E}(\theta; \delta) \leq \mathcal{L}_s(\theta) + \frac{1}{S} \sum_{i=1}^S \mathcal{S}_i(\theta; \rho).$$

*where $\rho(\delta)$ is defined in equation 11 of Appendix B.2. Moreover, there exists a model parameter $\theta$ such that*

$$\mathcal{E}(\theta; \delta) > \mathcal{L}_s(\theta) + \mathcal{S}_{agg}(\theta; \rho).$$

The proof is provided in Appendix B.2. Theorem 3.2 highlights that minimizing the average of per-domain sharpness provides a valid upper bound for our principled DG risk measure, $\mathcal{E}(\theta; \delta)$. In contrast, it also shows that aggregated sharpness offers no such guarantee, and can indeed be smaller even when the true risk is higher. This result confirms that minimizing per-domain sharpness is not merely an alternative, but a more appropriate and theoretically grounded surrogate for robust generalization under domain shifts.

# 4 METHODOLOGY

Our goal is to design an algorithm that effectively controls per-domain sharpness across all source domains, as motivated in Section 3. The conventional SAM approach, which perturbs parameters along the single, aggregated gradient of the total loss, is ill-suited for this task. The total gradient is often misaligned with domain-wise gradients, resulting in a suboptimal perturbation that fails to uniformly increase domain-specific losses. We provide a detailed analysis and empirical illustration of this failure mode in Appendix A. To overcome this limitation, in Section 4.1, we propose Domain-wise Gradual Sharpness-Aware Minimization (DGSAM) that employs a gradual, domain-specific perturbation mechanism to control per-domain sharpness. Subsequently, in Section 4.2, we provide a theoretical analysis of how this mechanism implicitly controls per-domain sharpness.

## 4.1 THE DGSAM ALGORITHM

DGSAM's update strategy is built upon a sequential perturbation scheme. Unlike the conventional SAM that uses a single perturbation, DGSAM sequentially incorporates the unique gradient from each source domain in successive steps. This transforms the perturbation process into a principled mechanism for integrating geometric information from multiple domains, allowing for more effective control of per-domain sharpness. The update rule of DGSAM is given by:

$$\theta_{t+1} = \theta_t - \gamma \left( \frac{S}{S+1} \right) \sum_{j=1}^{S+1} g_j, \quad \text{where} \tag{3}$$

$$g_j = \nabla \mathcal{L}_{B_{l_j}}(\tilde{\theta}_{j-1}) \text{ for } j = 1, \ldots, S, \quad g_{S+1} = \nabla \mathcal{L}_{B_{l_1}}(\tilde{\theta}_S). \tag{4}$$

where $l = (l_1, \ldots, l_S)$ denotes a random permutation of the $S$ source domain indices, and each $\mathcal{L}_{B_{l_j}}$ is the loss computed over a mini-batch $B_{l_j}$ drawn from the $l_j$-th domain.

In the ascent phase, as defined in equation 4, DGSAM performs $S + 1$ perturbation steps, each based on the gradient of a separate domain, followed by a descent step that updates the model using the aggregated gradients. Specifically, we begin with $\widetilde{\theta}_0 = \theta_t$ and at each step $j \in \{1, \ldots, S\}$, we compute the domain-specific gradient $g_j = \nabla \mathcal{L}_{B_{l_j}}(\widetilde{\theta}_{j-1})$ for the $j$-th domain (sampled in random order) and apply the perturbation $\rho \frac{g_j}{\|g_j\|}$ to update $\widetilde{\theta}_j$ (See lines 7-9 in Algorithm 1). These gradients are stored and later reused during the descent update to reduce computational overhead.

Note that the gradient $g_1$ is computed at the unperturbed point $\theta_t$ so it does not reflect the curvature-aware structure. To correct for this inconsistency, we perform one additional gradient computation at the final perturbed point $\widetilde{\theta}_S$ using $\nabla \mathcal{L}_{B_{l_1}}(\widetilde{\theta}_S)$ again (lines 10-11 in Algorithm 1). This ensures that all gradients contributing to the final update step are computed at perturbed points.

As a result, DGSAM collects $S + 1$ gradients along a trajectory that sequentially accounts for each domain's geometry. These gradients are then averaged for the final parameter update, as in equation 3. This design ensures that the descent direction is a more uniform reflection of all respective domain geometries, preventing the bias towards a single dominant domain that can occur with conventional SAM. Furthermore, this design is computationally efficient by reusing the gradients from the ascent phase, DGSAM requires only $S + 1$ gradient computations per iteration, compared to the $2S$ required by standard SAM.

The following theorem shows that DGSAM achieves $\epsilon$-stationarity under standard assumptions, aligning with the convergence guarantees recently established for SAM in non-convex settings Oikonomou & Loizou (2025).

---

**Algorithm 1** DGSAM

1: **Require:** Initial parameter $\theta_0$, learning rate $\gamma$, ; radius $\rho$; total iterations $N$; training sets $\{\mathcal{D}_i\}_{i=1}^S$
2: **for** $t \leftarrow 0$ to $N - 1$ **do**
3:      Sample batches $B_i \sim \mathcal{D}_i$ for $i = 1, \cdots, S$, and set a random order $l = permute(\{1, \cdots, S\})$
4:      $\tilde{\theta}_0 \leftarrow \theta_t$
5:      **for** $j \leftarrow 1$ to $S + 1$ **do**
6:          **if** $j \leq S$ **then**
7:              $g_j \leftarrow \nabla \mathcal{L}_{B_{l_j}}(\tilde{\theta}_{j-1})$
8:              $\tilde{\theta}_j \leftarrow \tilde{\theta}_{j-1} + \rho \frac{g_j}{\|g_j\|}$
9:          **else if** $j = S + 1$ **then**
10:             $g_{S+1} \leftarrow \nabla \mathcal{L}_{B_{l_1}}(\tilde{\theta}_S)$
11:          **end if**
12:      **end for**
13:      $\theta_{t+1} \leftarrow \theta_t - \gamma \left( \frac{S}{S+1} \right) \sum_{j=1}^{S+1} g_j$
14: **end for**

---

**Theorem 4.1** ($\epsilon$-approximate stationary). *Let Assumptions B.4 hold. Then, for any $\epsilon > 0$, the iterates of DGSAM satisfy for $\rho \leq \overline{\rho}$, $\gamma \leq \overline{\gamma}$, $T \geq \overline{T}$*

$$\min_{t=0,\ldots,T-1} \mathbb{E}\|\nabla \mathcal{L}_s(\theta_t)\| \leq \epsilon$$

*where full expressions of $\overline{\rho}$, $\overline{\gamma}$, and $\overline{T}$ are given in Theorem B.10. We refer to Appendix B.3 for the proof.*

## 4.2 HOW DGSAM CONTROLS PER-DOMAIN SHARPNESS

Recent studies (Ma et al., 2023; Zhuang et al., 2022) have pointed out that SAM's first-order approximations may lead to suboptimal control of curvature. Luo et al. (2024) showed that aligning the perturbation direction with an eigenvector can control the corresponding eigenvalue. However, relying solely on the top eigenvectors is insufficient in multi-domain settings, where the directions may conflict across domains. In such cases, it is more desirable to incorporate a broader set of eigenvectors associated with large eigenvalues, capturing curvature shared across domains. Moreover, Wen et al. (2023) demonstrated that controlling the entire eigenvalue spectrum yields tighter generalization bounds than focusing solely on the top eigenvalue.

In this regard, we analyze how DGSAM's gradual perturbation mechanism implicitly controls the per-domain sharpness. At the $j$-th step of the ascent phase, the gradient $g_j$ is computed as:

$$g_j = \nabla \mathcal{L}_{B_{l_j}}(\tilde{\theta}_{j-1}) = \nabla \mathcal{L}_{B_{l_j}} \left( \tilde{\theta}_0 + \sum_{k=1}^{j-1} \rho \frac{g_k}{\|g_k\|} \right)$$

$$\approx \nabla \mathcal{L}_{B_{l_j}}(\tilde{\theta}_0) + \rho \nabla^2 \mathcal{L}_{B_{l_j}}(\tilde{\theta}_0) \sum_{k=1}^{j-1} \frac{g_k}{\|g_k\|} + O(\rho^2).$$

Since the Hessian $\nabla^2 \mathcal{L}_{B_{l_j}}$ is symmetric and hence diagonalizable, we decompose it as $\nabla^2 \mathcal{L}_{B_{l_j}}(\tilde{\theta}_0) = \sum_n \lambda_n v_n v_n^\top$, where $E_j = \{(\lambda_n, v_n)\}$ is the set of eigenpairs of $\nabla^2 \mathcal{L}_{B_{l_j}}(\theta_t)$. Then, the $g_j$ can be approximated as

$$g_j \approx \nabla \mathcal{L}_{B_{l_j}}(\tilde{\theta}_0) + \rho \sum_{(\lambda,v) \in E_j} \lambda \left( \sum_{k=1}^{j-1} \frac{v^\top g_k}{\|v\|\|g_k\|} \right) v, \tag{5}$$

In this approximation, the first term represents the standard ascent direction for the $j$-th domain, while the second term is a curvature-aware correction term. This correction is a weighted sum of the Hessian's eigenvectors, where the weights depend on both the eigenvalues $\lambda$ and the alignment of eigenvectors with the perturbation directions from all previous domains ($g_1, \ldots, g_{j-1}$). Thus, DGSAM's gradual perturbation strategy naturally integrates curvature information from the entire sequence of domains, ensuring that the per-domain sharpness is controlled in a balanced and robust manner. This theoretical insight is confirmed empirically. In Appendix C.2, we show that the curvature-aware correction term contributes significantly to the ascent direction. Furthermore, this mechanism's effectiveness is demonstrated in our toy example (Section 3), where DGSAM consistently finds the truly flat minima and avoids the fake flat minima trap

## 5 NUMERICAL EXPERIMENTS

### 5.1 EXPERIMENTAL SETTINGS

**Evaluation protocols, Baselines and Datasets** For all main experiments, we adhere to the DomainBed protocol (Gulrajani & Lopez-Paz, 2021), including model initialization, hyperparameter tuning, and validation methods, to ensure a fair comparison. Our experiments are conducted on five widely used DG benchmarks: PACS (Li et al., 2017b), VLCS (Fang et al., 2013), OfficeHome (Venkateswara et al., 2017), TerraIncognita (Beery et al., 2018), and DomainNet (Peng et al., 2019).

We adopt the standard leave-one-domain-out setup: one domain is held out for testing, while the model is trained on the remaining source domains (Gulrajani & Lopez-Paz, 2021). Model selection is based on validation accuracy computed over the source domains. In addition to the average test accuracy commonly reported in DG, we also report the standard deviation of per-domain performance across test domains. This metric captures robustness to domain shifts and highlights potential overfitting to domains that are similar to the training distribution. Each experiment is repeated three times, and standard errors are reported.

**Implementation Details** We use a ResNet-50 (He et al., 2016) backbone pretrained on ImageNet, and Adam (Kingma & Ba, 2015) as the base optimizer. We use the hyperparameter space, the total number of iterations, and checkpoint frequency based on Wang et al. (2023). The specific hyperparameter settings and search ranges are described in Appendix F.1.

### 5.2 ACCURACY AND DOMAIN-WISE VARIANCE ACROSS BENCHMARKS

**Baselines on the DomainBed Protocol.** We compare DGSAM with 18 baseline algorithms across five widely used benchmark datasets: PACS, VLCS, OfficeHome, TerraIncognita, and DomainNet. The complete experimental setup and evaluation protocol follow DomainBed (Gulrajani & Lopez-Paz, 2021). Table 1 reports the average test accuracy and two types of standard deviation: (1) trial-based standard deviation across three random seeds, denoted by $\pm$, and (2) domain-wise standard deviation, measuring performance variance across held-out domains. Higher accuracy and lower

Table 1: Performance comparison on five DomainBed benchmarks. We report both trial-based standard deviation ($\pm$) and test-domain standard deviation (SD). Bold and underlined entries indicate the **best** and second-best results, excluding combined methods. Baseline results are sourced from prior work (see Appendix G for references).

| Algorithms | PACS Mean | SD | VLCS Mean | SD | OfficeHome Mean | SD | TerraInc Mean | SD | DomainNet Mean | SD | Avg Mean | SD | (s/iter) |
|---|---|---|---|---|---|---|---|---|---|---|---|---|---|
| ARM[†] | 85.1±0.6 | 8.0 | 77.6±0.7 | 13.1 | 64.8±0.4 | 10.2 | 45.5±1.3 | 7.4 | 35.5±0.5 | 16.7 | 61.7 | 11.1 | 0.12 |
| VREx[†] | 84.9±1.1 | 7.6 | 78.3±0.8 | 12.4 | 66.4±0.6 | 9.9 | 46.4±2.4 | 6.9 | 33.6±3.0 | 15.0 | 61.9 | 10.4 | 0.12 |
| RSC[†] | 85.2±1.0 | 7.6 | 77.1±0.7 | 13.0 | 65.5±1.0 | 10.0 | 46.6±1.0 | 7.0 | 38.9±0.7 | 17.3 | 62.7 | 11.0 | 0.15 |
| MTL[†] | 84.6±1.0 | 8.0 | 77.2±0.4 | 12.5 | 66.4±0.5 | 10.0 | 45.6±2.4 | 7.3 | 40.6±0.3 | 18.4 | 62.9 | 11.2 | 0.14 |
| ERM[†] | 85.5±0.6 | 7.0 | 77.3±1.1 | 12.5 | 66.5±0.4 | 10.8 | 46.1±2.9 | 8.0 | 40.9±0.3 | 18.6 | 63.3 | 11.4 | 0.12 |
| SagNet[†] | 86.3±0.5 | 6.9 | 77.8±0.7 | 12.5 | 68.1±0.3 | 9.5 | 48.6±0.3 | 7.1 | 40.3±0.3 | 17.9 | 64.2 | 10.8 | 0.36 |
| CORAL[†] | 86.2±0.6 | 7.5 | 78.8±0.7 | 12.0 | 68.7±0.4 | 9.6 | 47.7±0.4 | 7.0 | 41.5±0.3 | 18.3 | 64.6 | 10.9 | 0.14 |
| GGA | 86.4±1.7 | 6.6 | 78.7±1.0 | 12.2 | 67.0±0.5 | 10.5 | 48.5±2.0 | 7.4 | 44.5±0.3 | 19.7 | 65.0 | 11.3 | 0.54 |
| GGA-L | 86.5±1.5 | 6.6 | 78.4±1.0 | 12.6 | 66.5±0.4 | 10.0 | 49.8±2.8 | 6.0 | 44.5±0.3 | 19.7 | 65.1 | 11.0 | 0.36 |
| GENIE | 87.8±0.6 | 6.8 | 80.7±0.7 | 11.7 | 69.7±0.5 | 10.0 | 52.0±2.1 | 5.5 | 44.1±0.5 | 19.4 | 66.9 | 10.7 | 0.10 |
| SWAD | 88.1±0.4 | 5.9 | 79.1±0.4 | 12.8 | 70.6±0.3 | 9.2 | 50.0±0.3 | 7.9 | 46.5±0.2 | 19.9 | 66.9 | 11.2 | 0.12 |
| GAM[‡] | 86.1±1.3 | 7.4 | 78.5±1.2 | 12.5 | 68.2±0.8 | 12.8 | 45.2±1.7 | 9.1 | 43.8±0.3 | 20.0 | 64.4 | 12.4 | 0.49 |
| SAM[†] | 85.8±1.3 | 6.9 | 79.4±0.6 | 12.5 | 69.6±0.3 | 9.5 | 43.3±0.3 | 7.5 | 44.3±0.2 | 19.4 | 64.5 | 11.2 | 0.24 |
| Lookbehind-SAM | 86.0±0.4 | 7.2 | 78.9±0.8 | 12.4 | 69.2±0.6 | 11.2 | 44.5±1.0 | 8.2 | 44.2±0.3 | 19.6 | 64.7 | 11.8 | 0.54 |
| GSAM[†] | 85.9±0.3 | 7.4 | 79.1±0.3 | 12.3 | 69.3±0.1 | 9.9 | 47.0±0.1 | 8.8 | 44.6±0.3 | 19.8 | 65.2 | 11.6 | 0.25 |
| FAD | 88.2±0.6 | 6.3 | 78.9±0.9 | 12.1 | 69.2±0.7 | 13.4 | 45.7±1.6 | 9.6 | 44.4±0.3 | 19.5 | 65.3 | 12.2 | 0.42 |
| DISAM | 87.1±0.5 | 5.6 | 79.9±0.2 | 12.3 | 70.3±0.2 | 10.3 | 46.6±1.4 | 6.9 | 45.4±0.3 | 19.5 | 65.9 | 10.9 | 0.37 |
| SAGM | 86.6±0.3 | 7.2 | 80.0±0.4 | 12.3 | 70.1±0.3 | 9.4 | 48.8±0.3 | 7.5 | 45.0±0.2 | 19.4 | 66.1 | 11.2 | 0.24 |
| DGSAM | 88.5±0.4 | 5.2 | 81.4±0.5 | 11.5 | 70.8±0.3 | 8.5 | 50.4±0.7 | 6.9 | 45.5±0.3 | 19.4 | 67.3 | 10.3 | 0.19 |
| DGSAM + CORAL | 88.8±0.4 | 5.2 | 81.9±0.4 | 11.4 | 71.2±0.4 | 8.6 | 50.8±0.7 | 6.9 | 46.2±0.3 | 19.5 | 67.8 | 10.3 | 0.19 |
| DGSAM + SWAD | 88.7±0.4 | 5.4 | 80.9±0.5 | 11.6 | 71.4±0.4 | 8.7 | 51.1±0.8 | 6.8 | 47.1±0.3 | 19.6 | 67.8 | 10.4 | 0.19 |
| DGSAM + Mixup | 89.4±0.4 | 5.5 | 81.7±0.4 | 11.4 | 71.3±0.3 | 8.0 | 50.5±0.6 | 6.9 | 48.3±0.3 | 19.7 | 68.2 | 10.3 | 0.20 |
| DGSAM + ERM++ | 90.1±0.5 | 5.3 | 81.0±0.3 | 11.5 | 74.9±0.2 | 8.6 | 52.1±0.9 | 6.4 | 51.0±0.3 | 20.9 | 69.8 | 10.5 | 0.29 |

Table 2: DG performances on ViT-B/16 backbone.

| Algorithms | PACS | VLCS | OfficeHome | TerraInc | DomainNet | Avg |
|---|---|---|---|---|---|---|
| CORAL | 95.4 | 82.5 | 83.3 | 52.0 | 59.5 | 74.5 |
| DISAM | 96.8 | 82.2 | 84.2 | 51.4 | 59.9 | 74.9 |
| ERM | 96.6 | 80.9 | 84.1 | 55.5 | 59.2 | 75.3 |
| SAM | 96.1 | 83.5 | 86.5 | 61.2 | 60.5 | 76.3 |
| DGSAM | **97.3** | **84.5** | **87.3** | **62.2** | **78.5** | **77.8** |

standard deviation indicate better and more robust generalization. DGSAM achieves the highest average accuracy 67.3% and the lowest domain-level variance 10.3 among all methods, outperforming baselines on PACS, VLCS, and OfficeHome, and ranking second on TerraIncognita and DomainNet.

**Combination with Other DG Strategies.** Beyond its strong standalone performance, DGSAM also serves as a complementary component to other DG strategies. As shown in Table 1, integrating DGSAM with diverse and orthogonal methods, including SWAD, Mixup (Lopez-Paz et al., 2018), CORAL (Sun & Saenko, 2016), and ERM++ (Teterwak et al., 2025), consistently yields further performance gains. This demonstrates the broad applicability of DGSAM as a foundational optimizer that can enhance various DG frameworks. Detailed per-dataset results are provided in Appendix F.2.

**Performance on a Large-Scale Backbone (ViT-B/16).** While the standard DomainBed protocol provides a crucial benchmark, the ResNet-50 backbone is a relatively small-scale model. To demonstrate that DGSAM is effective and scalable for more realistic, large-scale architectures, we therefore conduct additional experiments using a Vision Transformer (ViT-B/16) backbone. As shown in Table 2, DGSAM again consistently outperforms strong baselines, underscoring its effectiveness across different architectures.

## 5.3 Sharpness Analysis

To verify that DGSAM effectively induces flatter minima, we analyze the geometry of the loss landscape at the converged model parameters using a ResNet-50 backbone on the DomainNet dataset. We report three sharpness metrics: zeroth-order sharpness, the trace of the loss Hessian estimated

Table 3: Comparison of the three sharpness metrics across different methods. **Sep.** denotes the average per-domain sharpness across separate source domains, where the value in parentheses represents the **domain-wise standard deviation**, i.e., variance across domains.

| Method | Zeroth-order Sharpness | | Hessian Trace | | Maximum Eigenvalue | |
|---|---|---|---|---|---|---|
| | Sep. Mean (SD) | Aggregated | Sep. Mean (SD) | Aggregated | Sep. Mean (SD) | Aggregated |
| ERM | 17.90 (5.62) | 34.06 | 940.52 (181.66) | 1372.51 | 89.24 (17.02) | 121.86 |
| SAM | 4.79 (2.17) | 19.68 | 5.83 (2.38) | 9.31 | 1.51 (0.77) | 1.85 |
| SAGM | 4.52 (2.34) | 12.38 | 2.49 (1.76) | **4.84** | 0.73 (0.36) | 1.23 |
| DISAM | 3.95 (1.83) | 8.14 | 3.50 (2.63) | 5.70 | 0.83 (0.29) | 1.45 |
| DGSAM | **2.98 (1.40)** | **6.41** | **2.13 (1.52)** | 4.93 | **0.65 (0.27)** | **1.18** |

via Hutchinson's method Ubaru et al. (2017); Avron & Toledo (2011), and its maximum eigenvalue ($\lambda_{max}$) computed using the Lanczos algorithm Ghorbani et al. (2019a); Lin et al. (2016). As shown in Table 3, DGSAM consistently outperforms the baselines. Notably, while SAGM exhibits a marginally lower aggregated Hessian trace, DGSAM achieves a lower mean and standard deviation in the per-domain Hessian trace. This empirically validates our theoretical analysis that minimizing per-domain sharpness is more critical for robust generalization than minimizing the aggregated average, as it ensures no specific domain remains sharp. This improved geometry is further corroborated by the Hessian spectrum density in Figure 3, where DGSAM effectively suppresses the spectral tail and controls the entire spectrum more effectively than SAM.

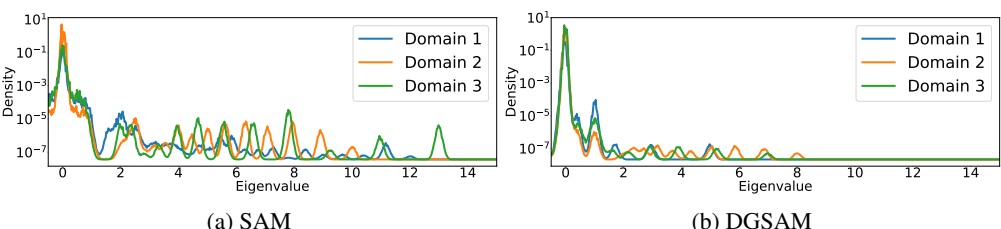

(a) SAM                                   (b) DGSAM

Figure 3: Hessian spectrum density at converged Minima: (a) SAM and (b) DGSAM.

### 5.4 COMPUTATIONAL COST

In addition to performance improvements, DGSAM significantly reduces the computational overhead commonly associated with SAM variants. Let $S$ denote the number of source domains and $c$ the unit cost of computing gradients for one mini-batch. Then, the per-iteration cost of ERM is $S \times c$, as it requires one gradient computation per domain. SAM performs two backpropagations for all domain, yielding a cost

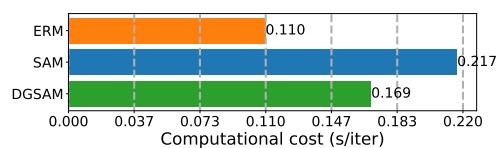

Figure 4: Comparison of empirical computational cost measured by training time per iteration.

of $2S \times c$. In contrast, DGSAM requires only $S + 1$ gradient computations, resulting in a theoretical cost of $(S + 1) \times c$. Further details are provided in the Appendix D.1.

To validate this, we measure the actual training time per iteration on the PACS dataset. With $S = 3$ source domains, ERM takes $S \times c = 0.11$ seconds per iteration. SAM incurs a cost of 0.217 seconds, nearly double that of ERM, while DGSAM achieves 0.169 seconds per iteration. Although slightly higher than its theoretical cost $(S + 1) \times c \approx 0.148$, the deviation is primarily due to additional overheads such as gradient aggregation. Moreover, this efficiency is not achieved at the expense of memory. As detailed in Appendix D.2, DGSAM requires less peak memory than both ERM and SAM. Full results of cost on all datasets are included in Appendix F.2.

## 6 DISCUSSION AND FUTURE DIRECTIONS

This paper revisits the role of sharpness minimization in domain generalization. While prior approaches have naively applied SAM to the aggregated loss across source domains, we reveal that this strategy can converge to *fake flat minima*—solutions that appear flat on total loss but remain sharp in separate domains, leading to poor generalization. To better capture the structure of domain-specific risks, we introduced a new perspective based on the *average worst-case domain risk*, showing that minimizing per-domain sharpness offers more meaningful control over robustness to distribution shift than minimizing aggregated sharpness. This insight offers a fundamentally new direction for the DG community, shifting the sharpness-aware optimization paradigm from single-source modeling to domain-specific objectives. Based on this finding, we proposed DGSAM, an algorithm that gradually applies perturbations along domain-specific directions and reuses gradients to efficiently reduce per-domain sharpness. Experiments on five DG benchmarks showed that DGSAM not only improves average accuracy but also significantly reduces domain-wise variance, achieving flatter minima across respective domains and better generalization to unseen distributions.

Our findings open a new direction for sharpness-aware domain generalization, but leave several open questions. When all local minima correspond to fake flat minima, it remains unclear which solutions are truly optimal or how to guide the model toward them. Developing a more direct method for minimizing per-domain sharpness, beyond sequential perturbation, could further improve training stability and theoretical guarantees. Finally, because SAM is widely applied in multi-loss settings such as multi-task learning (Le et al., 2024; Phan et al., 2022) and federated learning (Lee & Yoon, 2024; Qu et al., 2022; Caldarola et al., 2022), careful treatment of per-domain sharpness may likewise enhance generalization in these broader contexts.

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

# Appendix Contents

# Appendix

## A  LIMITATIONS OF TOTAL GRADIENT PERTURBATION

In SAM, each iteration performs gradient ascent to identify sensitive directions in the loss landscape by perturbing the parameters as

$$\tilde{\theta}_t = \theta_t + \epsilon^*_{\mathcal{D}_s} = \theta_t + \rho \frac{\nabla \mathcal{L}_s(\theta_t)}{\|\nabla \mathcal{L}_s(\theta_t)\|}, \tag{6}$$

where $\epsilon^*_{\mathcal{D}_s}$ is the perturbation computed from the total loss gradient. However, this update direction may not increase losses uniformly across source domains, as the total loss gradient $\nabla \mathcal{L}_s(\theta_t)$ does not generally align with the per-domain gradients $\nabla \mathcal{L}_i(\theta_t)$ for $i = 1, \ldots, S$, as discussed in Section 3).

This misalignment between the total gradient and per-domain gradients leads to suboptimal perturbations when applied uniformly across all domains. To empirically demonstrate this limitation, we visualize in Figure 5 how different perturbation strategies affect the domain-wise loss increments during training. Starting from $\theta_0$, we iteratively apply perturbations to compute the perturbed parameter $\tilde{\theta}_i = \theta_0 + \sum_{j=1}^{i} \epsilon_j$ on the DomainNet dataset (Peng et al., 2019) using ResNet-50 (He et al., 2016). In Figure 5a, each $\epsilon_i$ is computed using the total loss gradient. In contrast, Figure 5b applies perturbations sequentially using domain-specific gradients.

As shown in Figure 5a, total gradient perturbations often increase losses in an imbalanced manner across domains. On the other hand, the domain-wise perturbation strategy in Figure 5b leads to a more uniform increase in domain-wise losses. This observation suggests that applying domain-specific gradients sequentially is more effective at capturing the structure of per-domain losses. As a result, the resulting perturbations better reflect per-domain sharpness.

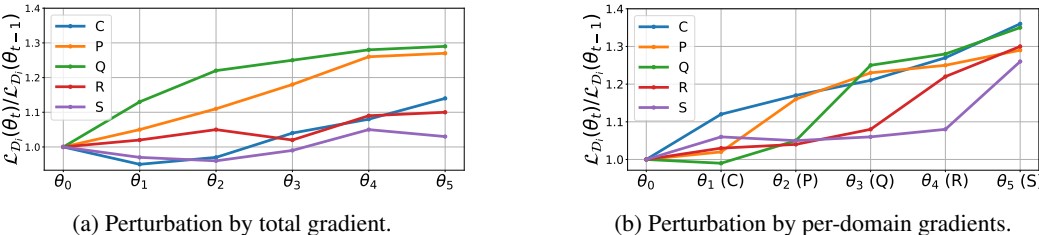

(a) Perturbation by total gradient.    (b) Perturbation by per-domain gradients.

Figure 5: Domain-wise loss increments under different perturbation strategies.

## B  THEORETICAL ANALYSIS AND PROOFS

### B.1  PROOF OF PROPOSITION 3.1

*Proof of Proposition 3.1.* Let $\theta$ be a strict local minimum such that $\nabla L_s(\theta) = 0$ and $H(\theta) = \nabla^2 L_s(\theta) \succ 0$. Suppose $\rho$ is sufficiently small. Then, the second-order Taylor expansion for $\mathcal{L}_s$ and $\mathcal{L}_i$ gives:

$$\mathcal{L}_s(\theta + \epsilon) = \mathcal{L}_s(\theta) + \nabla \mathcal{L}_s(\theta)^\top \epsilon + \frac{1}{2} \epsilon^\top H(\theta) \epsilon + o(\|\epsilon\|^2)$$

and

$$\mathcal{L}_i(\theta + \epsilon) = \mathcal{L}_i(\theta) + \nabla \mathcal{L}_i(\theta)^\top \epsilon + \frac{1}{2} \epsilon^\top H_i(\theta) \epsilon + o(\|\epsilon\|^2), \ i = 1, \ldots, S$$

where $H$ and $H_i$ are the Hessian matrices for $\mathcal{L}_s$ and $\mathcal{L}_i$, respectively, evaluated at $\theta$.

Then, using $\nabla \mathcal{L}_s(\theta) = 0$ and $H(\theta) = \frac{1}{S} \sum_{i=1}^{S} H_i(\theta)$, we have

$$\mathcal{L}_s(\theta + \epsilon) - \mathcal{L}_s(\theta) = \frac{1}{2} \epsilon^\top \left( \frac{1}{S} \sum_{i=1}^{S} H_i(\theta) \right) \epsilon + o(\|\epsilon\|^2)$$

which yields the zeroth-order sharpness for $\mathcal{L}_s$:

$$\mathcal{S}_{\text{agg}}(\theta; \rho) = \max_{\|\epsilon\| \leq \rho} (\mathcal{L}_s(\theta + \epsilon) - \mathcal{L}_s(\theta)) = \frac{1}{2S} \rho^2 \sigma_{max} \left( \sum_{i=1}^{S} H_i(\theta) \right) + o(\|\rho\|^2)$$

where $\sigma_{max}(A)$ denotes the largest eigenvalue of the matrix $A$.

To show that the statement does not hold in general, it suffices to provide a counterexample. First, we consider the case where $\|\nabla \mathcal{L}_i(\theta)\| = 0$ for all $i = 1, 2, \ldots, S$. Then, the zeroth-order sharpness of the $i$-th domain loss function is given by

$$\mathcal{S}_i(\theta; \rho) = \frac{1}{2} \rho^2 \sigma_{max} (H_i(\theta)) + o(\|\rho\|^2).$$

This leads to the following expression of the average sharpness over all per-domain loss functions:

$$\frac{1}{S} \sum_{i=1}^{S} \mathcal{S}_i(\theta; \rho) = \frac{1}{2S} \rho^2 \sum_{i=1}^{S} \sigma_{max} (H_i(\theta)) + o(\|\rho\|^2).$$

Next, consider two different local minima $\theta_1$ and $\theta_2$. For sufficiently small $\rho$, we can write:

$$\mathcal{S}_{\text{agg}}(\theta_1; \theta) < \mathcal{S}_{\text{agg}}(\theta_2; \rho) \tag{7}$$

$$\Leftrightarrow$$

$$\sigma_{max} \left( \sum_{i=1}^{S} H_i(\theta_1) \right) < \sigma_{max} \left( \sum_{i=1}^{S} H_i(\theta_2) \right). \tag{8}$$

Similarly, for sufficiently small $\rho$, we have the following relationship between the average per-domain sharpnesses at $\theta_1$ and $\theta_2$:

$$\frac{1}{S} \sum_{i=1}^{S} \mathcal{S}_i(\theta; \rho) < \frac{1}{S} \sum_{i=1}^{S} \mathcal{S}_i(\theta; \rho) \tag{9}$$

$$\Leftrightarrow$$

$$\sum_{i=1}^{S} \sigma_{max} (H_i(\theta_1)) < \sum_{i=1}^{S} \sigma_{max} (H_i(\theta_2)). \tag{10}$$

Consequently, we conclude that Equation 7 does not imply Equation 9 since the largest eigenvalue of a sum of matrices, $\sigma_{max} \left( \sum_{i=1}^{S} H_i(\theta) \right)$, is not generally equal to the sum of the largest eigenvalues of the per-domain matrices, $\sum_{i=1}^{S} \sigma_{max} (H_i(\theta))$.

Secondly, let us consider the case where $\nabla \mathcal{L}_s(\theta) = 0$, but there exists at least two elements such that $\nabla \mathcal{L}_i(\theta) \neq 0$. For simplicity, let $S = 2$. Without loss of generality, assume $\nabla \mathcal{L}_1(\theta) > 0$ and $\nabla \mathcal{L}_2(\theta) = -\nabla \mathcal{L}_1(\theta)$. Then, the sharpness for $\mathcal{L}_1(\theta)$ is given by

$$\mathcal{S}_1(\theta; \rho) = \|\nabla \mathcal{L}_1(\theta)\| \rho + o(\|\rho\|).$$

Now, consider two local minima $\theta_1$ and $\theta_2$ satisfying the following inequality:

$$\mathcal{S}_{\text{agg}}(\theta_1; \rho) < \mathcal{S}_{\text{agg}}(\theta_2; \rho).$$

A counterexample can be constructed such that for some $G > 0$ and $0 < c < 1$,

$$\nabla \mathcal{L}_1(\theta_1) = G = -\nabla \mathcal{L}_2(\theta_1),$$

and

$$\nabla \mathcal{L}_1(\theta_2) = cG = -\nabla \mathcal{L}_2(\theta_2).$$

In this example, we find that $\frac{1}{S} \sum_{i=1}^{S} \mathcal{S}_i(\theta_1; \rho) > \frac{1}{S} \sum_{i=1}^{S} \mathcal{S}_i(\theta_2; \rho)$,. However, such a choice of gradients does not affect the Hessian matrices, and thus the inequality for the sharpness of the total loss remains unchanged. Therefore, the sharpness for the total loss does not generally follow the same ordering as the average sharpness of the per-domain losses. $\square$

## B.2 PROOF OF THEOREM 3.2

We begin by imposing some standard conditions on the loss function.

**Assumption B.1.** For each $i$, let $\mathcal{D}_i$ be the $i$-th source domain distribution and $\mathcal{L}_{\mathcal{D}_i}(\theta) = \mathbb{E}_{X \sim \mathcal{D}_i}[\ell(\theta, X)]$ where $\ell$ is a loss function. Assume that $\ell(\theta, x)$ is uniformly bounded for all $\theta$ and $x$ and Lipschitz continuous in $\theta$. That is, there exist $M$ and $G$ such that

$$|\ell(\theta, x)| \leq M, \quad |\ell(\theta, x) - \ell(\theta', x)| \leq G\|\theta - \theta'\| \quad \text{for all } \theta, \theta', x.$$

Moreover, if $\mathrm{Div} = W_1$ (the Wasserstein-1 distance), assume additionally that for each $\theta$, the map $x \mapsto \ell(\theta, x)$ is $L_x$–Lipschitz, i.e.

$$|\ell(\theta, x) - \ell(\theta, x')| \leq L_x\, d(x, x') \quad \text{for all } \theta, \theta', x.$$

Under Assumption B.1, the following lemma states the relationship between distribution shifts and parameter perturbations.

**Lemma B.2.** *Let Assumption B.1 hold, and let $\mathcal{D}_i$ be the $i$th source distribution with*

$$\mathcal{L}_i(\theta) = \mathbb{E}_{x \sim \mathcal{D}_i}[\ell(\theta; x)].$$

*Fix a divergence or distance* $\mathrm{Div}$ *and threshold* $\delta > 0$*, and set*

$$\mathcal{U}_i^\delta = \{\, D : \mathrm{Div}(D\|\mathcal{D}_i) \leq \delta \}.$$

*Define the perturbation radius*

$$\rho(\delta) = \begin{cases} \dfrac{M}{G}\sqrt{\dfrac{\delta}{2}}, & \text{if } \mathrm{Div} = D_{\mathrm{KL}}, \\[2mm] \dfrac{M}{G}\,\delta, & \text{if } \mathrm{Div} = \|\cdot\|_{TV}, \\[2mm] \dfrac{L_x}{G}\,\delta, & \text{if } \mathrm{Div} = W_1. \end{cases} \tag{11}$$

*Then for all* $\theta$ *and any* $\rho \geq \rho(\delta)$*,*

$$\sup_{D \in \mathcal{U}_i^\delta} \mathcal{L}_D(\theta) \;\leq\; \max_{\|\epsilon\| \leq \rho} \mathcal{L}_i(\theta + \epsilon).$$

*Proof.* Fix $\rho \geq \rho(\delta)$ where

$$\rho(\delta) = \begin{cases} \dfrac{M}{G}\sqrt{\dfrac{\delta}{2}}, & \mathrm{Div} = D_{\mathrm{KL}}, \\[2mm] \dfrac{M}{G}\,\delta, & \mathrm{Div} = \|\cdot\|_{TV}, \\[2mm] \dfrac{L_x}{G}\,\delta, & \mathrm{Div} = W_1. \end{cases}$$

We will show in each case that for all $\mathcal{D}$ with $\mathrm{Div}(\mathcal{D}\|\mathcal{D}_i) \leq \delta$,

$$\big|\mathcal{L}_D(\theta) - \mathcal{L}_i(\theta)\big| \leq G\,\rho(\delta).$$

**Case (i):** $\mathrm{Div} = D_{\mathrm{KL}}$ and $\rho(\delta) = \frac{M}{G}\sqrt{\delta/2}$. Pinsker's inequality gives

$$\|\mathcal{D} - \mathcal{D}_i\|_{TV} \leq \sqrt{\tfrac{1}{2} D_{\mathrm{KL}}(\mathcal{D}\|\mathcal{D}_i)} \leq \sqrt{\tfrac{\delta}{2}},$$

which leads to

$$\big|\mathcal{L}_\mathcal{D}(\theta) - \mathcal{L}_i(\theta)\big| \leq M\,\|\mathcal{D} - \mathcal{D}_i\|_{TV} \leq M\sqrt{\tfrac{\delta}{2}} = G\,\rho(\delta).$$

**Case (ii):** $\mathrm{Div} = \|\cdot\|_{TV}$ and $\rho(\delta) = \frac{M}{G}\,\delta$. The definition of total variation directly yields

$$\big|\mathcal{L}_\mathcal{D}(\theta) - \mathcal{L}_i(\theta)\big| \leq M\|\mathcal{D} - \mathcal{D}_i\|_{TV} \leq M\delta = G\rho(\delta).$$

**Case (iii):** $\text{Div} = W_1$ and $\rho(\delta) = \frac{L_x}{G}\delta$. Assume in addition that $x \mapsto \ell(\theta; x)$ is $L_x$-Lipschitz. Then by the Kantorovich–Rubinstein duality, we have

$$\left|\mathcal{L}_{\mathcal{D}}(\theta) - \mathcal{L}_i(\theta)\right| \leq L_x\, W_1(\mathcal{D}, \mathcal{D}_i) \leq L_x\,\delta = G\rho(\delta).$$

In each case, therefore, we obtain for all $\mathcal{D} \in \mathcal{U}_i^\delta$

$$\mathcal{L}_{\mathcal{D}}(\theta) \leq \mathcal{L}_i(\theta) + G\rho \tag{12}$$

On the other hand, for any perturbation $\epsilon$ with $\|\epsilon\| \leq \rho$, using the Lipschitz continuity of $\ell(\cdot, x)$, we obtain

$$\mathcal{L}_i(\theta + \epsilon) - \mathcal{L}_i(\theta) = \mathbb{E}_{x \sim \mathcal{D}_i}\big[\ell(\theta + \epsilon, x) - \ell(\theta, x)\big] \leq G\|\epsilon\|$$

which yields

$$\max_{\|\epsilon\| \leq \rho} \mathcal{L}_i(\theta + \epsilon) \leq \mathcal{L}_i(\theta) + G\rho. \tag{13}$$

Combining equation 12 and equation 13 and then taking the supremum over $\mathcal{D} \in \mathcal{U}_i^\delta$ gives

$$\sup_{D \in \mathcal{U}_i^\delta} \mathcal{L}_D(\theta) \leq \max_{\|\epsilon\| \leq \rho} \mathcal{L}_{\mathcal{D}_i}(\theta + \epsilon).$$

$\square$

Now, we are ready to prove Theorem 3.2.

***Proof of Theorem 3.2.*** Recall that

$$\mathcal{E}(\theta; \delta) = \frac{1}{S}\sum_{i=1}^{S} \sup_{\mathcal{D} \in \mathcal{U}_i^\delta} \mathcal{L}_{\mathcal{D}}(\theta),$$

and

$$\mathcal{L}_s(\theta) = \frac{1}{S}\sum_{i=1}^{S} \mathcal{L}_i(\theta).$$

By Lemma B.2, for each $i$ and $\rho \geq \rho(\delta)$, we have

$$\sup_{D \in \mathcal{U}_i^\delta} \mathcal{L}_D(\theta) \leq \max_{\|\epsilon\| \leq \rho} \mathcal{L}_i(\theta + \epsilon) = \mathcal{L}_i(\theta) + S_i(\theta; \rho).$$

where $\mathcal{S}_i(\theta; \rho) = \max_{\|\epsilon\| \leq \rho} \mathcal{L}_i(\theta + \epsilon) - \mathcal{L}_i(\theta)$ is the per-domain sharpness for domain $i$. Averaging over $i = 1, \ldots, S$ directly gives

$$\mathcal{E}(\theta; \delta) = \frac{1}{S}\sum_{i=1}^{S} \sup_{D \in \mathcal{U}_i^\delta} \mathcal{L}_D(\theta)$$

$$\leq \frac{1}{S}\sum_{i=1}^{S}\big[\mathcal{L}_i(\theta) + S_i(\theta; \rho)\big]$$

$$= L_s(\theta) + \frac{1}{S}\sum_{i=1}^{S} S_i(\theta; \rho).$$

It remains to show that no analogous bound in terms of the aggregated sharpness $\mathcal{S}_{\text{agg}}(\theta; \rho)$ can hold uniformly. To this end, it is enough to find a counterexample. Let $S = 2$ and $\text{Div} = D_{\text{KL}}$. Fix the source distributions $\mathcal{D}_1 = \mathcal{D}_2 = \text{Uni}\{-1, +1\}$ and define $\ell(\theta, x) = \theta x, \theta \in [0, 1]$. Then, one can compute

$$\mathcal{L}_1(\theta) = \mathcal{L}_2(\theta) = \mathbb{E}_{X \sim \mathcal{D}_i}[\theta X] = 0, \quad L_s(\theta) = \frac{\mathcal{L}_1(\theta) + \mathcal{L}_2(\theta)}{2} = 0.$$

If we take $\delta = \ln 2$, the adversarial set $\mathcal{U}_i^\delta$ contains both point-masses $\delta_{+1}$ and $\delta_{-1}$. Hence, we have

$$\sup_{D \in \mathcal{U}_i^\delta} \mathcal{L}_D(\theta) = \max_{x \in \{+1, -1\}} \theta\, x = \theta,$$

and therefore $\mathcal{E}(\theta; \delta) = \theta$. On the other hand, the aggregated sharpness is trivially zero since $\mathcal{L}_s(\theta) = 0$. Thus for any $\theta$, we find

$$\mathcal{E}(\theta; \delta) = \theta > 0 = L_s(\theta) + S_{\mathrm{agg}}(\theta; \rho),$$

showing that no uniform bound of the form $\mathcal{E}(\theta; \delta) \leq \mathcal{L}_s(\theta) + \mathcal{S}_{\mathrm{agg}}(\theta; \rho)$ can hold.

$\square$

### B.3 CONVERGENCE ANALYSIS

Our convergence analysis builds upon the techniques developed in Gower et al. (2019); Khaled & Richtárik (2020); Oikonomou & Loizou (2025).

#### B.3.1 PRELIMINARIES

**Definition B.3** (Domain-wise Subsampling and Stochastic Gradient, (Gower et al., 2019; Khaled & Richtárik, 2020)). Let $\mathcal{D}_1, \ldots, \mathcal{D}_S$ be $S$ source domains, and $i$-th data point is associated with per-domain loss functions $\mathcal{L}^i(\theta)$, where $\theta \in \mathbb{R}^p$ denotes the model parameters. We define the total loss function as:

$$\mathcal{L}_{\mathrm{s}}(\theta) := \frac{1}{n} \sum_{i=1}^{n} \mathcal{L}^i(\theta),$$

where $n$ is the total number of training samples aggregated from all domains.

We consider a two-level sampling process: First, a domain index $r \in \{1, \ldots, S\}$ is selected uniformly at random. Then, a minibatch $B_r \subset \mathcal{D}_r$ of fixed size $\tau$ is sampled uniformly from within the selected domain. The domain-wise sampling vector $v^{\mathcal{Q}} = (v_1^{\mathcal{Q}}, \ldots, v_n^{\mathcal{Q}})$ is drawn from a distribution $\mathcal{Q}$ defined by this two-level process. For each sample $i$, the sampling weight is given by:

$$v_i^{\mathcal{Q}} := \frac{S \cdot 1_{i \in B_r}}{\tau},$$

where $1_{i \in B_r}$ is the indicator function that equals 1 if sample $i$ is included in the minibatch and 0 otherwise. The resulting domain-wise stochastic gradient estimator is:

$$g^{\mathcal{Q}}(\theta) := \sum_i v_i^{\mathcal{Q}} \nabla \mathcal{L}^{(i)}(\theta).$$

where $\mathcal{L}^{(i)}$ is the loss evaluated on the $i$-th sample. According to the general arbitrary sampling paradigm (Gower et al., 2019), since $v^{\mathcal{Q}} \sim \mathcal{Q}$ satisfies $\mathbb{E}[v_i^{\mathcal{Q}}] = 1$ for all $i$, the estimator $g^{\mathcal{Q}}(\theta)$ is unbiased:

$$\mathbb{E}_{\mathcal{Q}}[g^{\mathcal{Q}}(\theta)] = \nabla \mathcal{L}_{\mathrm{s}}(\theta).$$

Furthermore, the second moment $\mathbb{E}[\|v_i^{\mathcal{Q}}\|^2]$ is finite under this scheme.

**Assumption B.4.** Let $\mathcal{B}$ be a minibatch sampled from the domain-wise subsampling distribution the domain-wise subsampling distribution $\mathcal{Q}$ defined in Definition B.3, and let $\mathcal{L}_{\mathcal{B}}$ denote the loss evaluated on $\mathcal{B}$. We assume that $\mathcal{L}_{\mathcal{B}}$ is $L$-smooth. That is, there exists a constant $L > 0$ such that for all $\theta, \theta'$ and any $\mathcal{B}$,

$$\|\nabla \mathcal{L}_{\mathcal{B}}(\theta) - \nabla \mathcal{L}_{\mathcal{B}}(\theta')\| \leq L\|\theta - \theta'\|. \tag{14}$$

**Definition B.5** (Expected Residual Condition). Let $\theta^* = \arg\min_\theta \mathcal{L}_{\mathrm{s}}(\theta)$. We say the Expected Residual condition is satisfied if there exist nonnegative constants $M_1, M_2, M_3 \geq 0$ such that, for any point $\theta$, the following inequality holds for an unbiased estimator (stochastic gradient) $g(\theta)$ of the true gradient $\nabla \mathcal{L}_{\mathrm{s}}(\theta)$:

$$\mathbb{E}\|g(\theta)\|^2 \leq 2M_1[\mathcal{L}_{\mathrm{s}}(\theta) - \mathcal{L}_{\mathrm{s}}(\theta^*)] + M_2\|\nabla \mathcal{L}_{\mathrm{s}}(\theta)\|^2 + M_3.$$

**Corollary B.6.** *Let Assumption B.4 holds and let the domain-wise stochastic gradient by $g^{\mathcal{Q}}(\theta)$ which is an unbiased estimator of $\mathcal{L}_s(\theta)$ for all $\theta$ with $\mathbb{E}[\|v_i^{\mathcal{Q}}\|^2] \leq \infty$. Then, it holds that*

$$\mathbb{E}_{\mathcal{Q}}\|g^{\mathcal{Q}}(\theta)\|^2 \leq 2M_1[\mathcal{L}_s(\theta) - \mathcal{L}_s(\theta^*)] + M_2\|\nabla \mathcal{L}_s(\theta)\|^2 + M_3.$$

*Proof.* In Proposition 2 of Khaled & Richtárik (2020), it is proved that $L$-smoothness and unbiased stochastic gradient with $\mathbb{E}_{\mathcal{D}}[v_i^2] < \infty$ imply Expected Residual condition (Definition B.5). $\square$

We collect a few basic inequalities that are frequently used throughout the proofs: For any $a, b \in \mathbb{R}^d$ and any $\beta > 0$, we have:

$$|\langle a, b\rangle| \leq \frac{1}{2\beta}\|a\|^2 + \frac{\beta}{2}\|b\|^2, \tag{15}$$

$$\|a + b\|^2 \leq (1 + \beta^{-1})\|a\|^2 + (1 + \beta)\|b\|^2, \tag{16}$$

$$\|a + b\|^2 \leq 2\|a\|^2 + 2\|b\|^2, \tag{17}$$

$$\left\|\sum_{i=1}^{n} x_i\right\|^2 \leq n\sum_{i=1}^{n}\|x_i\|^2. \tag{18}$$

### B.3.2 LEMMAS

We use a uniformly random permutation $\{l_1, \ldots, l_S\}$ over the domain indices. $B_{l_j}$ means minibatch from j-th chosen domain and the choice of order is initialized at every step. Thus $B_{l_j}$ is the domain-wise subsampling with definition B.3. For notational simplicity, we will write $g_j^t = \nabla\mathcal{L}_{B_{l_j}}\left(\theta_t + \sum_{k=1}^{j-1}\rho\frac{g_k^t}{\|g_k^t\|}\right)$.

**Lemma B.7.** *Let Assumption B.4 hold. Then the following inequality holds:*

$$\mathbb{E}_{\mathcal{Q}}\|g_j^t\|^2 \leq 2S^2L^2\rho^2 + 2\mathbb{E}_{\mathcal{Q}}\|g^{\mathcal{Q}}(\theta_t)\|^2,$$

*where $S$ is the number of domains.*

*Proof.* It follows that

$$\mathbb{E}_{\mathcal{Q}}\|g_j^t\|^2 = \mathbb{E}_{\mathcal{Q}}\left\|\nabla\mathcal{L}_{B_{l_j}}\left(\theta_t + \sum_{k=1}^{j-1}\rho\frac{g_k^t}{\|g_k^t\|}\right)\right\|^2$$

$$= \mathbb{E}_{\mathcal{Q}}\left\|\nabla\mathcal{L}_{B_{l_j}}\left(\theta_t + \sum_{k=1}^{j-1}\rho\frac{g_k^t}{\|g_k^t\|}\right) - \nabla\mathcal{L}_{B_{l_j}}(\theta_t) + \nabla\mathcal{L}_{B_{l_j}}(\theta_t)\right\|^2$$

$$\overset{(17)}{\leq} 2\mathbb{E}_{\mathcal{Q}}\left\|\nabla\mathcal{L}_{B_{l_j}}\left(\theta_t + \sum_{k=1}^{j-1}\rho\frac{g_k^t}{\|g_k^t\|}\right) - \nabla\mathcal{L}_{B_{l_j}}(\theta_t)\right\|^2 + 2\mathbb{E}_{\mathcal{Q}}\left\|\nabla\mathcal{L}_{B_{l_j}}(\theta_t)\right\|^2$$

$$\overset{(14)}{\leq} 2L^2\rho^2\mathbb{E}_{\mathcal{Q}}\left\|\sum_{k=1}^{j-1}\frac{g_k^t}{\|g_k^t\|}\right\|^2 + 2\mathbb{E}_{\mathcal{Q}}\|g^{\mathcal{Q}}(\theta_t)\|^2$$

$$\overset{(18)}{\leq} 2L^2\rho^2(j-1)\sum_{k=1}^{j-1}\mathbb{E}_{\mathcal{Q}}\left\|\frac{g_k^t}{\|g_k^t\|}\right\|^2 + 2\mathbb{E}_{\mathcal{Q}}\|g^{\mathcal{Q}}(\theta_t)\|^2$$

$$\leq 2S^2L^2\rho^2 + 2\mathbb{E}_{\mathcal{Q}}\|g^{\mathcal{Q}}(\theta_t)\|^2.$$

$\square$

**Lemma B.8.** *Let Assumption B.4 hold. Then the following inequality holds:*

$$\mathbb{E}_{\mathcal{Q}}\langle g_j^t, \nabla\mathcal{L}_s(\theta_t)\rangle \geq -SL\rho + (1 - \frac{SL\rho}{4})\|\nabla\mathcal{L}_s(\theta_t)\|^2,$$

*where $S$ is the number of domains.*

*Proof.*

$$\mathbb{E}_{\mathcal{Q}}\langle g_j^t,\, \nabla\mathcal{L}_{\mathrm{s}}(\theta_t)\rangle = \mathbb{E}_{\mathcal{Q}}\left\langle \nabla\mathcal{L}_{B_{l_j}}\left(\theta_t + \sum_{k=1}^{j-1}\rho\frac{g_k^t}{\|g_k^t\|}\right),\, \nabla\mathcal{L}_{\mathrm{s}}(\theta_t)\right\rangle$$

$$= \mathbb{E}_{\mathcal{Q}}\left\langle \nabla\mathcal{L}_{B_{l_j}}\left(\theta_t + \sum_{k=1}^{j-1}\rho\frac{g_k^t}{\|g_k^t\|}\right) - \nabla\mathcal{L}_{B_{l_j}}(\theta_t),\, \nabla\mathcal{L}_{\mathrm{s}}(\theta_t)\right\rangle$$

$$+ \mathbb{E}_{\mathcal{Q}}\left\langle \nabla\mathcal{L}_{B_{l_j}}(\theta_t),\, \nabla\mathcal{L}_{\mathrm{s}}(\theta_t)\right\rangle.$$

We have

$$\mathbb{E}_{\mathcal{Q}}\left\langle \nabla\mathcal{L}_{B_{l_j}}(\theta_t),\, \nabla\mathcal{L}_{\mathrm{s}}(\theta_t)\right\rangle = \left\langle \mathbb{E}_{\mathcal{Q}}[\nabla\mathcal{L}_{B_{l_j}}(\theta_t)],\, \nabla\mathcal{L}_{\mathrm{s}}(\theta_t)\right\rangle$$

$$= \left\langle \mathbb{E}_{\mathcal{Q}}[g^{\mathcal{Q}}(\theta_t)],\, \nabla\mathcal{L}_{\mathrm{s}}(\theta_t)\right\rangle$$

$$= \|\nabla\mathcal{L}_{\mathrm{s}}(\theta_t)\|^2,$$

and for $\beta > 0$

$$-\mathbb{E}_{\mathcal{Q}}\left\langle \nabla\mathcal{L}_{B_{l_j}}\left(\theta_t + \sum_{k=1}^{j-1}\rho\frac{g_k^t}{\|g_k^t\|}\right) - \nabla\mathcal{L}_{B_{l_j}}(\theta_t),\, \nabla\mathcal{L}_{\mathrm{s}}(\theta_t)\right\rangle$$

$$\overset{(15)}{\leq} \frac{1}{2\beta}\mathbb{E}_{\mathcal{Q}}\left\|\nabla\mathcal{L}_{B_{l_j}}\left(\theta_t + \sum_{k=1}^{j-1}\rho\frac{g_k^t}{\|g_k^t\|}\right) - \nabla\mathcal{L}_{B_{l_j}}(\theta_t)\right\|^2 + \frac{\beta}{2}\mathbb{E}_{\mathcal{Q}}\|\nabla\mathcal{L}_{\mathrm{s}}(\theta_t)\|^2$$

$$\overset{(14)}{\leq} \frac{L^2\rho^2}{2\beta}\mathbb{E}_{\mathcal{Q}}\left\|\sum_{k=1}^{j-1}\frac{g_k^t}{\|g_k^t\|}\right\|^2 + \frac{\beta}{2}\|\nabla\mathcal{L}_{\mathrm{s}}(\theta_t)\|^2$$

$$\leq \frac{S^2L^2\rho^2}{2\beta} + \frac{\beta}{2}\|\nabla\mathcal{L}_{\mathrm{s}}(\theta_t)\|^2.$$

In sum,

$$\mathbb{E}_{\mathcal{Q}}\langle g_j^t,\, \nabla\mathcal{L}_{\mathrm{s}}(\theta_t)\rangle \geq -\frac{S^2L^2\rho^2}{2\beta} - \frac{\beta}{2}\|\nabla\mathcal{L}_{\mathrm{s}}(\theta_t)\|^2 + \|\nabla\mathcal{L}_{\mathrm{s}}(\theta_t)\|^2$$

$$= -\frac{S^2L^2\rho^2}{2\beta} + (1 - \frac{\beta}{2})\|\nabla\mathcal{L}_{\mathrm{s}}(\theta_t)\|^2$$

$$= -SL\rho + (1 - \frac{SL\rho}{4})\|\nabla\mathcal{L}_{\mathrm{s}}(\theta_t)\|^2$$

with $\beta = \frac{SL\rho}{2}$. $\qquad\square$

**Lemma B.9** (Lemma A.8, (Oikonomou & Loizou, 2025)). *Let $(r_t)_{t\geq 0}$ and $(\delta_t)_{t\geq 0}$ be sequences of non-negative real numbers and let $g > 1$ and $N \geq 0$. Assume that the following recursive relationship holds:*

$$r_t \leq g\delta_t - \delta_{t+1} + N \tag{19}$$

*Then it holds*

$$\min_{0\leq t\leq T-1} r_t \leq \frac{g^T}{T}\delta_0 + N.$$

### B.3.3   PROOF OF THEOREM

**Theorem B.10** ($\epsilon$-approximate stationary). *Let Assumption B.4 hold. Define*

$$T_{\min} = \frac{12M_4}{\epsilon^2 S}\max\{1, \frac{24M_1M_4SL}{\epsilon^2}, 4M_2L, 12M_3SL\},$$

$$\overline{\rho} = \frac{1}{SL}\min\{1, \frac{\epsilon^2}{12}, \frac{\epsilon}{2\sqrt{6L}}\},$$

$$\overline{\gamma} = \min\{1, \frac{1}{S\sqrt{2M_1LT}}, \frac{1}{4M_2L}, \frac{\epsilon^2}{12M_3SL}\}.$$

*For all $\epsilon > 0$, if the DGSAM iteration(3) is employed, then for $\rho \leq \overline{\rho}$, $\gamma \leq \overline{\gamma}$, $T \geq T_{\min}$*

$$\min_{t=0,\ldots,T-1} \mathbb{E}\|\nabla\mathcal{L}_s(\theta_t)\| \leq \epsilon$$

*where the initial optimality gap $M_4 = \mathcal{L}_s(\theta_0) - \mathcal{L}_s(\theta^*)$, $S$ is the number of domains, $M_1, M_2, M_3$ are the constants for the expected residual condition.*

*Proof.* For simplicity, we assume that the effect of the batch size is absorbed into the learning rate $\gamma$, i.e., $\gamma$ is defined as the product of the base learning rate and the batch size.

From the $L$-smoothness of $\mathcal{L}_s$, we have

$$\mathcal{L}_s(\theta_{t+1}) \leq \mathcal{L}_s(\theta_t) + \langle \nabla\mathcal{L}_s(\theta_t), \theta_{t+1} - \theta_t \rangle + \frac{L}{2}\|\theta_{t+1} - \theta_t\|^2$$

$$= \mathcal{L}_s(\theta_t) - \gamma\frac{S}{S+1}\left\langle \nabla\mathcal{L}_s(\theta_t), \sum_{j=1}^{S+1} g_j^t \right\rangle + \frac{L\gamma^2}{2}\left(\frac{S}{S+1}\right)^2\left\|\sum_{j=1}^{S+1} g_j^t\right\|^2,$$

since the DGSAM update is defined as $\theta_{t+1} = \theta_t - \gamma\frac{S}{S+1}\sum_{j=1}^{S+1} g_j^t$.

By taking the expectation,

$$\mathbb{E}_{\mathcal{Q}}[\mathcal{L}_s(\theta_{t+1}) - \mathcal{L}_s(\theta^*) \mid \theta_t] - [\mathcal{L}_s(\theta_t) - \mathcal{L}_s(\theta^*)]$$

$$\leq -\gamma\frac{S}{S+1}\sum_{j=1}^{S+1}\mathbb{E}_{\mathcal{Q}}\langle \nabla\mathcal{L}_s(\theta_t), g_j^t \rangle + \frac{L\gamma^2}{2}\left(\frac{S}{S+1}\right)^2\mathbb{E}_{\mathcal{Q}}\left\|\sum_{j=1}^{S+1} g_j^t\right\|^2$$

$$\overset{(18)}{\leq} -\gamma S\mathbb{E}_{\mathcal{Q}}\langle \nabla\mathcal{L}_s(\theta_t), g_j^t \rangle + \frac{L\gamma^2 S^2}{2}\mathbb{E}_{\mathcal{Q}}\|g_j^t\|^2$$

$$\overset{Lem.B.7,B.8}{\leq} -\gamma S\left(-SL\rho + (1 - \frac{SL\rho}{4})\|\nabla\mathcal{L}_s(\theta_t)\|^2\right) + \frac{L\gamma^2 S^2}{2}\left(2S^2L^2\rho^2 + 2\mathbb{E}_{\mathcal{Q}}\|g^{\mathcal{Q}}(\theta_t)\|^2\right)$$

$$= -S\gamma\left(1 - \frac{SL\rho}{4}\right)\|\nabla\mathcal{L}_s(\theta_t)\|^2 + LS^2\gamma^2\mathbb{E}_{\mathcal{Q}}\|g^{\mathcal{Q}}(\theta_t)\|^2 + S^2L\gamma\rho(1 + S^2L^2\gamma\rho)$$

$$\overset{Cor.B.6}{\leq} -S\gamma\left(1 - \frac{SL\rho}{4}\right)\|\nabla\mathcal{L}_s(\theta_t)\|^2 + 2M_1LS^2\gamma^2[\mathcal{L}_s(\theta_t) - \mathcal{L}_s(\theta^*)] + M_2LS\gamma^2\|\nabla\mathcal{L}_s(\theta_t)\|^2$$

$$+ M_3LS^2\gamma^2 + S^2L\gamma\rho(1 + S^2L^2\gamma\rho)$$

$$= -S\gamma\left(1 - \frac{SL\rho}{4} - M_2L\gamma\right)\|\nabla\mathcal{L}_s(\theta_t)\|^2 + 2M_1LS^2\gamma^2[\mathcal{L}_s(\theta_t) - \mathcal{L}_s(\theta^*)] + S^2L\gamma(\rho + S^2L^2\gamma\rho^2 + M_3\gamma)$$

$$\leq -\frac{S\gamma}{2}\|\nabla\mathcal{L}_s(\theta_t)\|^2 + 2M_1LS^2\gamma^2[\mathcal{L}_s(\theta_t) - \mathcal{L}_s(\theta^*)] + S^2L\gamma(\rho + S^2L^2\gamma\rho^2 + M_3\gamma).$$

The final inequality follows from the inequality $1 - \frac{SL\rho}{4} - M_2L\gamma \geq \frac{1}{2}$, which is obtained from our assumptions $\rho \leq \frac{1}{SL}$ and $\gamma \leq \frac{1}{4M_2L}$.

In sum,

$$\mathbb{E}_{\mathcal{D}}[\mathcal{L}_s(\theta_{t+1}) - \mathcal{L}_s(\theta^*)] - [\mathcal{L}_s(\theta_t) - \mathcal{L}_s(\theta^*)]$$

$$\leq -\frac{S\gamma}{2}\|\nabla\mathcal{L}_s(\theta_t)\|^2 + 2M_1LS^2\gamma^2[\mathcal{L}_s(\theta_t) - \mathcal{L}_s(\theta^*)] + S^2L\gamma(\rho + S^2L^2\gamma\rho^2 + M_3\gamma)$$

$$\implies \frac{S\gamma}{2}\|\nabla\mathcal{L}_s(\theta_t)\|^2 \leq (1 + 2M_1LS^2\gamma^2)[\mathcal{L}_s(\theta_t) - \mathcal{L}_s(\theta^*)] - \mathbb{E}_{\mathcal{D}}[\mathcal{L}_s(\theta_{t+1}) - \mathcal{L}_s(\theta^*)]$$

$$+ S^2L\gamma(\rho + S^2L^2\gamma\rho^2 + M_3\gamma). \tag{20}$$

By taking expectation and applying the tower property, we can conclude that

$$\mathbb{E}\|\nabla\mathcal{L}_s(\theta_t)\|^2 \leq (1 + 2M_1LS^2\gamma^2)\frac{2}{S\gamma}\mathbb{E}[\mathcal{L}_s(\theta_t) - \mathcal{L}_s(\theta^*)] - \frac{2}{S\gamma}\mathbb{E}[\mathcal{L}_s(\theta_{t+1}) - \mathcal{L}_s(\theta^*)]$$

$$+ 2SL(\rho + S^2L^2\gamma\rho^2 + M_3\gamma). \tag{21}$$

We now define the following auxiliary quantities:

$$r_t := \mathbb{E}\|\nabla\mathcal{L}_s(\theta_t)\|^2 \geq 0,$$

$$\delta_t := \frac{2}{S\gamma}\mathbb{E}[\mathcal{L}_s(\theta_t) - \mathcal{L}_s(\theta^*)] \geq 0,$$

$$g := (1 + 2M_1LS^2\gamma^2) > 1,$$

$$N := 2SL(\rho + S^2L^2\gamma\rho^2 + M_3\gamma).$$

With these definitions, inequality 21 becomes:

$$r_t \leq g\delta_t - \delta_{t+1} + N.$$

By applying Lemma B.9, we have

$$\min_{t=0,\ldots,T-1}\mathbb{E}\|\nabla\mathcal{L}_s(\theta_t)\|^2 \leq \frac{2(1 + 2M_1LS^2\gamma^2)^T}{TS\gamma}[\mathcal{L}_s(\theta_0) - \mathcal{L}_s(\theta^*)] + 2SL(\rho + S^2L^2\gamma\rho^2 + M_3\gamma).$$

From $1 + x \leq e^x$, we can get

$$(1 + 2M_1LS^2\gamma^2)^T \leq \exp(2TM_1LS^2\gamma^2) \leq \exp(1) \leq 3,$$

since we have $\gamma \leq \frac{1}{S\sqrt{2M_1LT}}$ which imply $2TM_1LS^2\gamma^2 \leq 1$.

Therefore,

$$\min_{t=0,\ldots,T-1}\mathbb{E}\|\nabla\mathcal{L}_s(\theta_t)\|^2 \leq \frac{6M_4}{TS\gamma} + 2SL(\rho + S^2L^2\gamma\rho^2 + M_3\gamma).$$

The second term is less than $\frac{\epsilon^2}{2}$ with assumptions:

$$2SL\rho \leq \frac{\epsilon^2}{6} \iff \rho \leq \frac{\epsilon^2}{12SL},$$

$$\gamma \leq 1,$$

$$4S^2L^3\gamma\rho^2 \leq \frac{\epsilon^2}{6} \iff \rho \leq \frac{\epsilon}{2SL\sqrt{6L}} \quad \text{with } \gamma \leq 1,$$

$$2SLM_3\gamma \leq \frac{\epsilon^2}{6} \iff \gamma \leq \frac{\epsilon^2}{12SLM_3}.$$

Likewise, we have the inequality for the first term:

$$\frac{6M_4}{TS\gamma} \leq \frac{\epsilon^2}{2} \iff T \geq \frac{12M_4}{\epsilon^2 S\gamma} \tag{22}$$

We have so far imposed the following inequalities on $\gamma$:

$$\gamma \leq \min\left\{\frac{1}{4M_2L}, \frac{1}{S\sqrt{2M_1LT}}, 1, \frac{\epsilon^2}{12M_3SL}\right\}$$

Consequently, $T$ must satisfy the following conditions for (22).

$$T \geq \max\left\{\frac{48M_2M_4L}{\epsilon^2 S}, \frac{288M_1M_4^2L}{\epsilon^4}, \frac{12M_4}{\epsilon^2 S}, \frac{144M_3M_4L}{\epsilon^2}\right\}$$

Finally, we have:

$$\min_{t=0,\ldots,T-1}\mathbb{E}\|\nabla\mathcal{L}_s(\theta_t)\|^2 \leq \epsilon^2.$$

withe these assumptions:

$$T \geq \frac{12M_4}{\epsilon^2 S}\max\{1, \frac{24M_1M_4SL}{\epsilon^2}, 4M_2L, 12M_3SL\},$$

$$\rho \leq \frac{1}{SL}\min\{1, \frac{\epsilon^2}{12}, \frac{\epsilon}{2\sqrt{6L}}\},$$

$$\gamma \leq \min\{1, \frac{1}{S\sqrt{2M_1LT}}, \frac{1}{4M_2L}, \frac{\epsilon^2}{12M_3SL}\}.$$

$\square$

## C ADDITIONAL EXPERIMENTS

### C.1 SENSITIVITY ANALYSIS OF DGSAM WITH RESPECT TO $\rho$

To analyze the sensitivity of DGSAM to $\rho$, we evaluated the performance of SAM and DGSAM across different $\rho$ values $\{0.001, 0.005, 0.01, 0.05, 0.1, 0.2\}$ on the PACS and TerraIncognita datasets. As shown in Figure 6, DGSAM consistently outperformed SAM and demonstrated superior performance over a wider range of $\rho$ values.

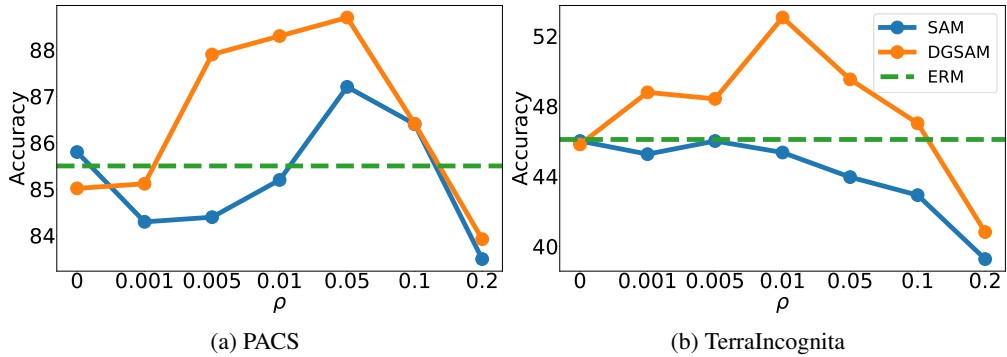

(a) PACS

(b) TerraIncognita

Figure 6: Sensitivity analysis

### C.2 COMPARISON OF TWO TERMS IN EQ 5

Figure 7 shows that the second term tends to be slightly smaller than the first term, but the two are comparable in magnitude. This indicates that both terms contribute to the gradual perturbation.

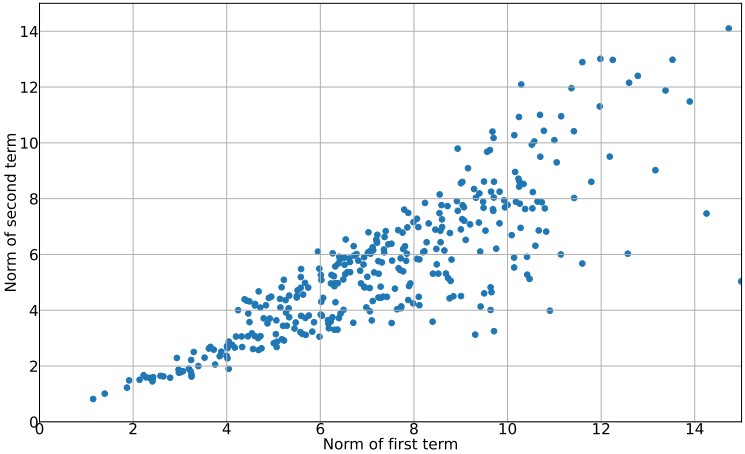

Figure 7: Comparison of magnitude of two terms in Eq 5 on the PACS

### C.3 ROBUSTNESS TO EXTREME DOMAIN IMBALANCE

To further validate the robustness of our proposed DGSAM method against domain imbalance, we conducted additional stress-test experiments under more extreme imbalance scenarios. For this analysis, we utilized the TerraIncognita dataset and artificially increased the sample size imbalance ratio between the largest and smallest domains from the original approximate ratio of 2:1 to 3:1, 5:1, and 10:1.

The results are presented in Table 4. As the domain imbalance becomes more severe, the performance of all methods gradually decreases. However, DGSAM consistently and significantly outperforms

both ERM and SAM across all tested scenarios. Notably, even with a severe 10:1 imbalance ratio, DGSAM's performance degrades gracefully while maintaining a substantial performance margin over the baselines. This result strongly demonstrates that DGSAM is inherently robust to domain heterogeneity and imbalance, owing to its mechanism of applying perturbations based on the normalized gradient for each domain.

Table 4: Performance comparison on TerraIncognita under varying degrees of domain imbalance.

| Method \ Ratio | 2:1 (Original) | 3:1 | 4:1 | 5:1 | 10:1 |
|---|---|---|---|---|---|
| ERM | 35.7 | 35.3 | 35.2 | 34.9 | 32.1 |
| SAM | 34.5 | 34.7 | 34.2 | 34.1 | 31.9 |
| **DGSAM** | **41.8** | **41.6** | **41.4** | **41.1** | **38.3** |

## C.4 SCALABILITY TO A LARGE NUMBER OF DOMAINS

The standard DGSAM implementation performs a sequential ascent over all $S$ source domains, which can become computationally inefficient and potentially unstable as the number of domains $S$ becomes very large. To address this scalability concern, we introduce a straightforward and practical modification: domain subsampling.

Instead of iterating through all $S$ domains, we can fix the number of sequential ascent steps to $k$ (where $k \ll S$, e.g., $k = 5$) by randomly subsampling a subset of $k$ domains at each training iteration. The method presented in the main manuscript is a specific case of this more general framework where $k = S$.

To verify the effectiveness of this approach, we applied DGSAM with domain subsampling ($k = 5$) to datasets comprising several tens of domains: PovertyMap (Yeh et al., 2020) and GlobalWheat (David et al., 2020). As shown in Table 5, DGSAM with subsampling not only addresses the scalability issue but also maintains strong performance, outperforming both ERM and SAM. This refinement confirms that DGSAM can be effectively and practically applied to large-scale scenarios with numerous domains.

Table 5: Performance on datasets with a large number of domains using domain subsampling.

| Method | PovertyMap (23 domains) | GlobalWheat (47 domains) |
|---|---|---|
| ERM | 0.45 | 50.8 |
| SAM | 0.44 | 51.1 |
| **DGSAM ($k = 5$ subsampling)** | **0.50** | **51.9** |

## C.5 ABLATION STUDIES ON STOCHASTIC ORDERING AND GRADIENT RE-USING

In this subsection, we empirically validate two critical design choices in the DGSAM algorithm: (1) the stochasticity in the sequential domain order, and (2) the gradient reuse strategy for computational efficiency. We conduct these ablation studies on the PACS and TerraIncognita datasets using the ResNet-50 backbone. The results are summarized in Table 6.

Table 6: Ablation analysis on PACS and TerraIncognita datasets.

| Method Configuration | PACS | | TerraIncognita | | s/iter |
|---|---|---|---|---|---|
| | Mean | (SD) | Mean | (SD) | |
| Not re-using | **88.9** | 0.5 | **51.3** | 0.5 | 0.236 |
| Fixed Order | 83.6 | 2.6 | 46.1 | 1.8 | 0.169 |
| DGSAM | 88.5 | 0.4 | 49.9 | 0.7 | 0.169 |

**Impact of Random Domain Permutation.** DGSAM permutes the order of source domains at each iteration before applying sequential perturbations. To assess the impact of this stochasticity, we compare our default setting with a "Fixed Order" variant, where the domain sequence for the gradual ascent remains constant throughout training. As presented in Table 6, fixing the domain order leads to a consistent degradation in average accuracy across benchmarks compared to the random permutation strategy. Furthermore, we observe a marked increase in performance variance, suggesting that a fixed sequence induces training instability. These results indicate that randomizing the perturbation order serves as an essential regularizer, preventing the optimization from biasing towards a specific trajectory and ensuring robust flatness across all domains.

**Effect of Gradient Re-using.** To minimize computational overhead, DGSAM approximates the descent direction by aggregating the gradients computed during the gradual ascent steps, rather than performing a fresh gradient computation at the final perturbed model parameter. We evaluate the trade-off of this design by comparing it with a variant that performs an additional backward pass at the final perturbed point to compute the exact gradient for the update. As shown in Table 6, while the additional gradient computation yields marginal gains in accuracy, it incurs a substantial computational penalty, leading to a considerable slowdown in training speed. Consequently, we adopt the gradient reuse strategy as the default, as it maintains competitive performance while significantly reducing the computational burden, offering a better balance for scalable domain generalization.

## C.6 DETAILS OF THE EXPERIMENTAL VERIFICATION OF SHARPNESS

## D COMPUTATION EFFICIENCY

### D.1 ILLUSTRATION OF COMPUTATIONAL COST COMPARISON

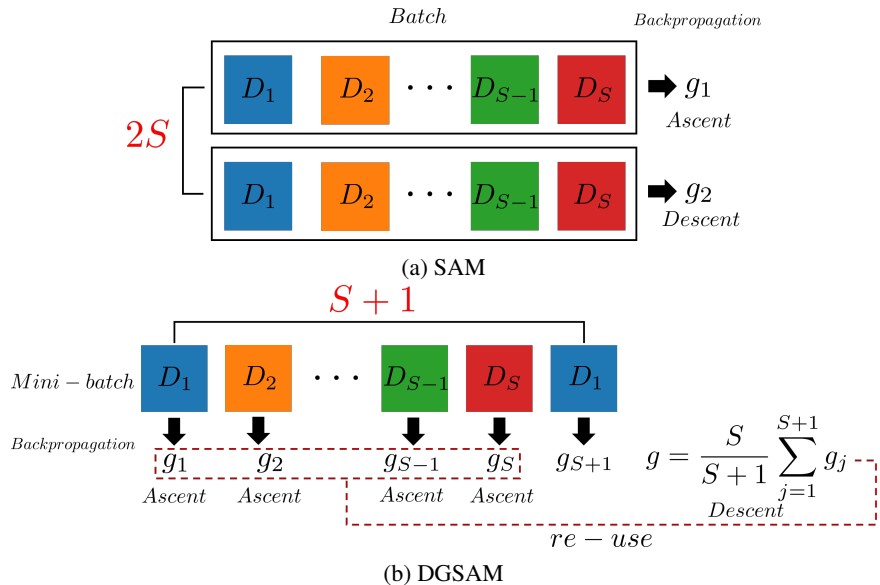

Figure 8: Computational cost of SAM and DGSAM.

In standard domain generalization tasks, a single update step operates on a batch that comprises mini-batches from all source domains. While the number of data samples per domain-specific mini-batch may vary, we follow the DomainBed protocol (Gulrajani & Lopez-Paz, 2021), where each mini-batch contains an equal number of samples. Throughout this paper, we assume uniform mini-batch sizes across domains.

Let the computational cost of computing the loss and performing backpropagation on a single domain-specific mini-batch from one domain be denoted as $c$. In the standard SAM algorithm, both an ascent and a descent gradient must be computed for each of the $S$ domain-specific mini-batches, resulting in a total gradient computation cost of $2S \times c$ per update theoretically.

In contrast, as illustrated in the Figure 8, DGSAM computes gradients separately for each mini-batch, using $g_1, \ldots, g_S$ not only as ascent gradients but also directly for the parameter update. Due to this efficient reuse of gradients, DGSAM requires only $(S + 1) \times c$ in gradient computation cost per update theoretically.

## D.2 ADDITIONAL ANALYSIS ON COMPUTATIONAL RESOURCES

We provide a comprehensive analysis of computational resources, including both computational complexity (GFLOPs) and memory usage. All measurements were conducted using a ResNet-50 backbone, and the results reported in Table 7 are averaged across the PACS and TerraIncognita datasets. We report GFLOPs per update alongside mean and maximum memory allocation.

**Computational Cost (GFLOPs).** We measure the GFLOPs required for a single model update. As expected, SAM nearly doubles the cost of ERM due to its dual forward-backward passes. DGSAM successfully reduces this overhead, validating our efficiency analysis.

**Memory Efficiency.** Despite the moderate increase in GFLOPs compared to ERM, DGSAM achieves the lowest memory consumption. While ERM and SAM typically perform the backward pass over a full batch including data from all domains, DGSAM performs backward passes separately on each domain-specific mini-batch, accumulating gradients before a single update. This approach prevents memory cost from scaling linearly with the number of domains, resulting in significantly lower memory usage compared to both ERM and SAM.

Table 7: Comparison of computational cost (GFLOPs per sample) and memory consumption (GB).

| Method | Computational Cost | Memory Usage | |
|---|---|---|---|
| | GFLOPs / sample | Mean (GB) | Max (GB) |
| ERM | 8.27 | 8.0 | 8.1 |
| SAM | 15.99 | 8.1 | 8.3 |
| DGSAM | **13.28** | **5.8** | **6.0** |

## E VISUALIZATION OF LOSS LANDSCAPES

Figure 9 shows the 3D loss landscapes of converged solutions obtained by SAM and our proposed DGSAM on the PACS dataset using ResNet-50. Each subplot corresponds to a different domain or the aggregated total loss. While SAM finds flat minima in the total loss, it fails to flatten the loss surfaces in respective domains. In contrast, DGSAM successfully reduces per-domain sharpness as well as the total sharpness, demonstrating its ability to achieve flatter minima at the domain level.

Figure 10 illustrates how DGSAM sequentially applies domain-specific perturbations and aggregates gradients to update the model.

## F DETAILS OF MAIN EXPERIMENTS

### F.1 IMPLEMENTATION DETAILS

We searched hyperparameters in the following ranges: the learning rate was chosen from $\{10^{-5}, 2 \times 10^{-5}, 3 \times 10^{-5}, 5 \times 10^{-5}\}$, the dropout rate from $\{0.0, 0.2, 0.5\}$, the weight decay from $\{10^{-4}, 10^{-6}\}$, and $\rho$ from $\{0.03, 0.05, 0.1\}$. Each experiment was repeated three times, using 20 randomly initialized models sampled from this space, following the DomainBed protocol (Gulrajani & Lopez-Paz, 2021). The optimal hyperparameters selected based on DomainBed criteria for each dataset are provided in Table 8 to ensure replicability. All our experiments were conducted on an NVIDIA A100 GPU, using Python 3.11.5, PyTorch 2.0.0, Torchvision 0.15.1, and CUDA 11.7.

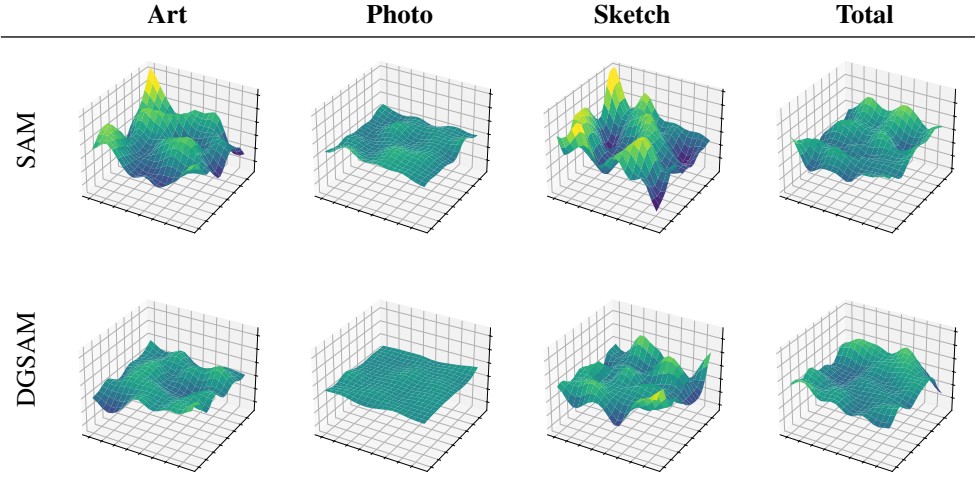

Figure 9: Comparison of loss landscapes of converged minima using SAM and DGSAM across different domains on the PACS dataset. We set the grid with two random direction. DGSAM performs better than SAM in reducing per-domain sharpness in all three respective domains, and total sharpness.

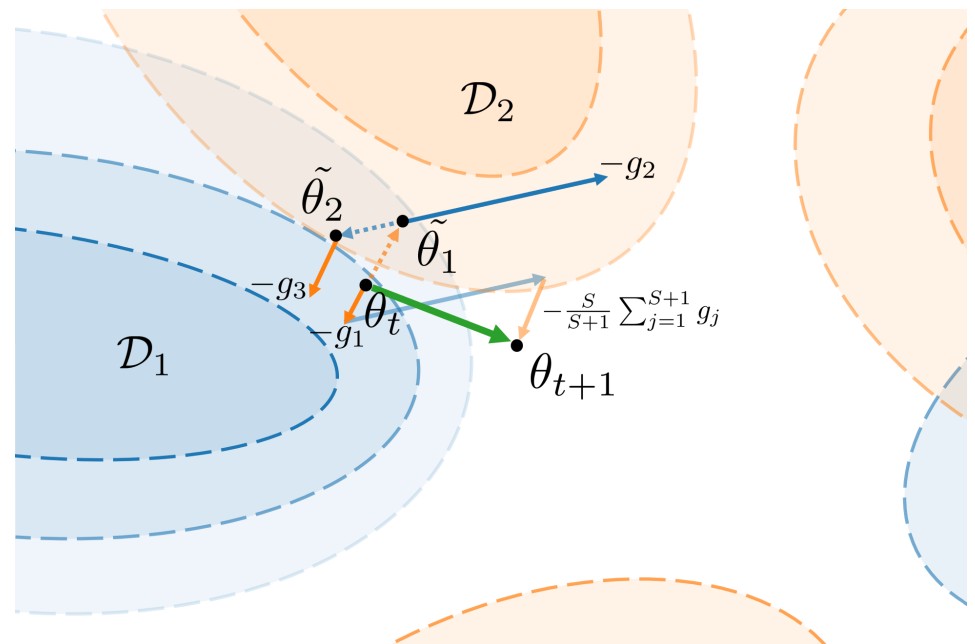

Figure 10: A visualization of DGSAM algorithm.

Table 8: Optimal hyperparameter settings for each dataset

| Dataset | Learning Rate | Dropout Rate | Weight Decay | $\rho$ |
|---|---|---|---|---|
| PACS | $3 \times 10^{-5}$ | 0.5 | $10^{-4}$ | 0.03 |
| VLCS | $10^{-5}$ | 0.5 | $10^{-4}$ | 0.03 |
| OfficeHome | $10^{-5}$ | 0.5 | $10^{-6}$ | 0.1 |
| TerraIncognita | $10^{-5}$ | 0.2 | $10^{-6}$ | 0.05 |
| DomainNet | $2 \times 10^{-5}$ | 0.5 | $10^{-4}$ | 0.1 |

## F.2 FULL RESULTS

Here are the detailed results of the main experiment in Section 5.2 for each dataset. The outcomes are marked with † if sourced from Wang et al. (2023), ‡ if sourced from Zhang et al. (2023a), and are unlabeled if sourced from individual papers. We note that all results were conducted in the same experimental settings as described in their respective papers. The value shown next to the performance for each test domain represents the standard error across three trials.

Table 9: The performance of DGSAM with 18 baseline algorithms on PACS.

| Algorithm | A | C | P | S | Avg | SD | (s/iter) |
|---|---|---|---|---|---|---|---|
| MTL† (Blanchard et al., 2021) | 87.5±0.8 | 77.1±0.5 | 96.4±0.8 | 77.3±1.8 | 84.6 | 8.0 | 0.12 |
| VREx† (Krueger et al., 2021) | 86.0±1.6 | 79.1±0.6 | 96.9±0.5 | 77.7±1.7 | 84.9 | 7.6 | 0.11 |
| ARM† (Zhang et al., 2021) | 86.8±0.6 | 76.8±0.5 | 97.4±0.3 | 79.3±1.2 | 85.1 | 8.0 | 0.11 |
| RSC† (Huang et al., 2020) | 85.4±0.8 | 79.7±1.8 | 97.6±0.3 | 78.2±1.2 | 85.2 | 7.6 | 0.14 |
| ERM† | 84.7±0.4 | 80.8±0.6 | 97.2±0.3 | 79.3±1.0 | 85.5 | 7.0 | 0.11 |
| CORAL† (Sun & Saenko, 2016) | 88.3±0.2 | 80.0±0.5 | 97.5±0.3 | 78.8±1.3 | 86.2 | 7.5 | 0.12 |
| SagNet† (Nam et al., 2021) | 87.4±1.0 | 80.7±0.6 | 97.1±0.1 | 80.0±0.4 | 86.3 | 6.9 | 0.32 |
| GGA (Ballas & Diou, 2025) | 86.5±1.8 | 81.2±3.0 | 97.1±0.9 | 80.8±0.9 | 86.4 | 6.6 | 0.49 |
| GGA-L (Ballas & Diou, 2025) | 88.0±1.0 | 81.2±2.0 | 97.1±0.3 | 80.8±2.5 | 86.5 | 6.6 | 0.33 |
| GENIE (Cho et al., 2025) | 88.7±0.7 | 82.8±1.3 | 98.5±0.1 | 81.3±0.4 | 87.8 | 6.8 | 0.09 |
| SWAD (Cha et al., 2021) | 89.3±0.2 | 83.4±0.6 | 97.3±0.3 | 82.5±0.5 | 88.1 | 5.9 | 0.11 |
| SAM† (Foret et al., 2021) | 85.6±2.1 | 80.9±1.2 | 97.0±0.4 | 79.6±1.6 | 85.8 | 6.9 | 0.22 |
| GSAM† (Zhuang et al., 2022) | 86.9±0.1 | 80.4±0.2 | 97.5±0.0 | 78.7±0.8 | 85.9 | 7.4 | 0.22 |
| Lookbehind-SAM (Mordido et al., 2024) | 86.8±0.2 | 80.2±0.3 | 97.4±0.8 | 79.7±0.2 | 86.0 | 7.2 | 0.50 |
| GAM‡ (Zhang et al., 2023b) | 85.9±0.9 | 81.3±1.6 | 98.2±0.4 | 79.0±2.1 | 86.1 | 7.4 | 0.43 |
| SAGM (Wang et al., 2023) | 87.4±0.2 | 80.2±0.3 | 98.0±0.2 | 80.8±0.6 | 86.6 | 7.2 | 0.22 |
| DISAM (Zhang et al., 2024) | 87.1±0.4 | 81.9±0.5 | 96.2±0.3 | 83.1±0.7 | 87.1 | 5.6 | 0.33 |
| FAD (Zhang et al., 2023a) | 88.5±0.5 | 83.0±0.8 | 98.4±0.2 | 82.8±0.9 | 88.2 | 6.3 | 0.38 |
| DGSAM | 88.9±0.2 | 84.8±0.7 | 96.9±0.2 | 83.5±0.3 | 88.5 | 5.2 | 0.17 |
| DGSAM + SWAD | 89.1±0.5 | 84.6±0.4 | 97.3±0.1 | 83.6±0.4 | 88.7 | 5.4 | 0.17 |
| DGSAM + CORAL | 89.5±0.3 | 84.9±0.3 | 97.0±0.2 | 83.7±0.7 | 88.8 | 5.2 | 0.18 |
| DGSAM + Mixup | 90.1±0.4 | 84.8±0.4 | 98.2±0.3 | 84.5±0.5 | 89.4 | 5.5 | 0.17 |
| DGSAM + ERM++ | 90.6±0.5 | 85.2±0.6 | 98.5±0.3 | 86.0±0.4 | 90.1 | 5.3 | 0.25 |

Table 10: The performance of DGSAM with 18 baseline algorithms on VLCS

| Algorithm | C | L | S | V | Avg | SD | (s/iter) |
|---|---|---|---|---|---|---|---|
| RSC† (Huang et al., 2020) | 97.9±0.1 | 62.5±0.7 | 72.3±1.2 | 75.6±0.8 | 77.1 | 13.0 | 0.13 |
| MTL† (Blanchard et al., 2021) | 97.8±0.4 | 64.3±0.3 | 71.5±0.7 | 75.3±1.7 | 77.2 | 12.5 | 0.12 |
| ERM† | 98.0±0.3 | 64.7±1.2 | 71.4±1.2 | 75.2±1.6 | 77.3 | 12.5 | 0.11 |
| ARM† (Zhang et al., 2021) | 98.7±0.2 | 63.6±0.7 | 71.3±1.2 | 76.7±0.6 | 77.6 | 13.1 | 0.11 |
| SagNet† (Nam et al., 2021) | 97.9±0.4 | 64.5±0.5 | 71.4±1.3 | 77.5±0.5 | 77.8 | 12.5 | 0.32 |
| VREx† (Krueger et al., 2021) | 98.4±0.3 | 64.4±1.4 | 74.1±0.4 | 76.2±1.3 | 78.3 | 12.4 | 0.11 |
| GGA-L (Ballas & Diou, 2025) | 98.9±0.4 | 66.5±0.3 | 70.0±2.0 | 78.1±1.1 | 78.4 | 12.6 | 0.33 |
| GGA (Ballas & Diou, 2025) | 98.4±0.2 | 65.4±0.1 | 73.8±1.6 | 77.4±1.9 | 78.7 | 12.2 | 0.49 |
| CORAL† (Sun & Saenko, 2016) | 98.3±0.1 | 66.1±1.2 | 73.4±0.3 | 77.5±1.2 | 78.8 | 12.0 | 0.12 |
| SWAD (Cha et al., 2021) | 98.8±0.1 | 63.3±0.3 | 75.3±0.5 | 79.2±0.6 | 79.1 | 12.8 | 0.11 |
| GENIE (Cho et al., 2025) | 99.3±0.3 | 67.2±1.5 | 76.6±0.3 | 79.7±0.8 | 80.7 | 11.7 | 0.09 |
| GAM‡ (Zhang et al., 2023b) | 98.8±0.6 | 65.1±1.2 | 72.9±1.0 | 77.2±1.9 | 78.5 | 12.5 | 0.43 |
| Lookbehind-SAM (Mordido et al., 2024) | 98.7±0.6 | 65.1±1.1 | 73.1±0.4 | 78.7±0.9 | 78.9 | 12.4 | 0.50 |
| FAD (Zhang et al., 2023a) | 99.1±0.5 | 66.8±0.9 | 73.6±1.0 | 76.1±1.3 | 78.9 | 12.1 | 0.38 |
| GSAM† (Zhuang et al., 2022) | 98.7±0.3 | 64.9±0.2 | 74.3±0.0 | 78.5±0.8 | 79.1 | 12.3 | 0.22 |
| SAM† (Foret et al., 2021) | 99.1±0.2 | 65.0±1.0 | 73.7±1.0 | 79.8±0.1 | 79.4 | 12.5 | 0.22 |
| DISAM (Zhang et al., 2024) | 99.3±0.0 | 66.3±0.5 | 81.0±0.1 | 73.2±0.1 | 79.9 | 12.3 | 0.33 |
| SAGM (Wang et al., 2023) | 99.0±0.2 | 65.2±0.4 | 75.1±0.3 | 80.7±0.8 | 80.0 | 12.3 | 0.22 |
| DGSAM + SWAD | 99.3±0.7 | 67.2±0.3 | 77.7±0.6 | 79.2±0.5 | 80.9 | 11.6 | 0.17 |
| DGSAM + ERM++ | 99.2±0.3 | 67.4±0.2 | 77.8±0.1 | 79.5±0.4 | 81.0 | 11.5 | 0.25 |
| DGSAM | 99.0±0.5 | 67.0±0.5 | 77.9±0.5 | 81.8±0.4 | 81.4 | 11.5 | 0.17 |
| DGSAM + Mixup | 99.1±0.4 | 67.3±0.5 | 78.1±0.2 | 82.1±0.5 | 81.7 | 11.4 | 0.17 |
| DGSAM + CORAL | 99.3±0.8 | 67.4±0.7 | 79.5±0.5 | 81.5±0.1 | 81.9 | 11.4 | 0.18 |

Table 11: The performance of DGSAM with 18 baseline algorithms on OfficeHome

| Algorithm | A | C | P | R | Avg | SD | (s/iter) |
|---|---|---|---|---|---|---|---|
| ARM[†] (Zhang et al., 2021) | 58.9±0.8 | 51.0±0.5 | 74.1±0.1 | 75.2±0.3 | 64.8 | 10.2 | 0.11 |
| RSC[†] (Huang et al., 2020) | 60.7±1.4 | 51.4±0.3 | 74.8±1.1 | 75.1±1.3 | 65.5 | 10.0 | 0.14 |
| MTL[†] (Blanchard et al., 2021) | 61.5±0.7 | 52.4±0.6 | 74.9±0.4 | 76.8±0.4 | 66.4 | 10.0 | 0.12 |
| VREx[†] (Krueger et al., 2021) | 60.7±0.9 | 53.0±0.9 | 75.3±0.1 | 76.6±0.5 | 66.4 | 9.9 | 0.11 |
| GGA-L (Ballas & Diou, 2025) | 59.7±0.2 | 53.8±0.5 | 75.3±0.8 | 77.1±0.1 | 66.5 | 10.0 | 0.33 |
| GGA (Ballas & Diou, 2025) | 61.7±0.1 | 52.5±0.5 | 77.1±1.3 | 77.0±0.1 | 67.0 | 10.5 | 0.49 |
| ERM[†] | 63.1±0.3 | 51.9±0.4 | 77.2±0.5 | 78.1±0.2 | 67.6 | 10.8 | 0.11 |
| SagNet[†] (Nam et al., 2021) | 63.4±0.2 | 54.8±0.4 | 75.8±0.4 | 78.3±0.3 | 68.1 | 9.5 | 0.32 |
| CORAL[†] (Sun & Saenko, 2016) | 65.3±0.4 | 54.4±0.5 | 76.5±0.1 | 78.4±0.5 | 68.7 | 9.6 | 0.12 |
| GENIE (Cho et al., 2025) | 66.2±0.5 | 55.0±0.4 | 77.5±0.4 | 80.0±0.5 | 69.7 | 10.0 | 0.09 |
| SWAD (Cha et al., 2021) | 66.1±0.4 | 57.7±0.4 | 78.4±0.1 | 80.2±0.2 | 70.6 | 9.2 | 0.11 |
| GAM[‡] (Zhang et al., 2023b) | 63.0±1.2 | 49.8±0.5 | 77.6±0.6 | 82.4±1.0 | 68.2 | 12.8 | 0.43 |
| FAD (Zhang et al., 2023a) | 63.5±1.0 | 50.3±0.8 | 78.0±0.4 | 85.0±0.6 | 69.2 | 13.4 | 0.40 |
| Lookbehind-SAM (Mordido et al., 2024) | 64.7±0.3 | 53.1±0.8 | 77.4±0.5 | 81.7±0.7 | 69.2 | 11.2 | 0.50 |
| GSAM[†] (Zhuang et al., 2022) | 64.9±0.1 | 55.2±0.2 | 77.8±0.0 | 79.2±0.0 | 69.3 | 9.9 | 0.22 |
| SAM[†] (Foret et al., 2021) | 64.5±0.3 | 56.5±0.2 | 77.4±0.1 | 79.8±0.4 | 69.6 | 9.5 | 0.22 |
| SAGM (Wang et al., 2023) | 65.4±0.4 | 57.0±0.3 | 78.0±0.3 | 80.0±0.2 | 70.1 | 9.4 | 0.22 |
| DISAM (Zhang et al., 2024) | 65.8±0.2 | 55.6±0.2 | 79.2±0.2 | 80.6±0.1 | 70.3 | 10.3 | 0.33 |
| DGSAM | 65.6±0.4 | 59.7±0.2 | 78.0±0.2 | 80.1±0.4 | 70.8 | 8.5 | 0.17 |
| DGSAM + CORAL | 66.4±0.5 | 59.6±0.2 | 78.3±0.3 | 80.5±0.5 | 71.2 | 8.6 | 0.18 |
| DGSAM + Mixup | 67.3±0.3 | 60.2±0.4 | 77.4±0.3 | 80.3±0.3 | 71.3 | 8.0 | 0.17 |
| DGSAM + SWAD | 66.2±0.6 | 59.9±0.1 | 78.1±0.4 | 81.2±0.5 | 71.4 | 8.7 | 0.17 |
| DGSAM + ERM++ | 70.9±0.5 | 62.7±0.1 | 82.3±0.2 | 83.8±0.1 | 74.9 | 8.6 | 0.25 |

Table 12: The performance of DGSAM with 18 baseline algorithms on TerraIncognita

| Algorithm | L100 | L38 | L43 | L46 | Avg | SD | (s/iter) |
|---|---|---|---|---|---|---|---|
| ARM[†] (Zhang et al., 2021) | 49.3±0.7 | 38.3±2.4 | 55.8±0.8 | 38.7±1.3 | 45.5 | 7.4 | 0.11 |
| MTL[†] (Blanchard et al., 2021) | 49.3±1.2 | 39.6±6.3 | 55.6±1.1 | 37.8±0.8 | 45.6 | 7.3 | 0.12 |
| ERM[†] | 49.8±4.4 | 42.1±1.4 | 56.9±1.8 | 35.7±3.9 | 46.1 | 8.0 | 0.11 |
| VREx[†] (Krueger et al., 2021) | 48.2±4.3 | 41.7±1.3 | 56.8±0.8 | 38.7±3.1 | 46.4 | 6.9 | 0.11 |
| RSC[†] (Huang et al., 2020) | 50.2±2.2 | 39.2±1.4 | 56.3±1.4 | 40.8±0.6 | 46.6 | 7.0 | 0.13 |
| CORAL[†] (Sun & Saenko, 2016) | 51.6±2.4 | 42.2±1.0 | 57.0±1.0 | 39.8±2.9 | 47.7 | 7.0 | 0.12 |
| GGA (Ballas & Diou, 2025) | 50.9±2.2 | 42.5±1.0 | 59.7±1.4 | 41.5±3.5 | 48.5 | 7.4 | 0.49 |
| SagNet[†] (Nam et al., 2021) | 53.0±2.9 | 43.0±2.5 | 57.9±0.6 | 40.4±1.3 | 48.6 | 7.1 | 0.32 |
| GGA-L (Ballas & Diou, 2025) | 57.2±5.2 | 45.1±1.0 | 56.4±1.4 | 44.5±3.5 | 49.8 | 6.0 | 0.33 |
| SWAD (Cha et al., 2021) | 55.4±0.0 | 44.9±1.1 | 59.7±0.4 | 39.9±0.2 | 50.0 | 7.9 | 0.11 |
| GENIE (Cho et al., 2025) | 55.2±4.8 | 47.5±2.1 | 59.2±0.4 | 45.9±1.0 | 52.0 | 5.5 | 0.09 |
| SAM[†] (Foret et al., 2021) | 46.3±1.0 | 38.4±2.4 | 54.0±1.0 | 34.5±0.8 | 43.3 | 7.5 | 0.22 |
| Lookbehind-SAM (Mordido et al., 2024) | 44.6±0.8 | 41.1±1.4 | 57.4±1.2 | 34.9±0.6 | 44.5 | 8.2 | 0.50 |
| GAM[‡] (Zhang et al., 2023b) | 42.2±2.6 | 42.9±1.7 | 60.2±1.8 | 35.5±0.7 | 45.2 | 9.1 | 0.43 |
| FAD (Zhang et al., 2023a) | 44.3±2.2 | 43.5±1.7 | 60.9±2.0 | 34.1±0.5 | 45.7 | 9.6 | 0.38 |
| DISAM (Zhang et al., 2024) | 46.2±2.9 | 41.6±0.1 | 58.0±0.5 | 40.5±2.2 | 46.6 | 6.9 | 0.33 |
| GSAM[†] (Zhuang et al., 2022) | 50.8±0.1 | 39.3±0.2 | 59.6±0.0 | 38.2±0.8 | 47.0 | 8.8 | 0.22 |
| SAGM (Wang et al., 2023) | 54.8±1.3 | 41.4±0.8 | 57.7±0.6 | 41.3±0.4 | 48.8 | 7.5 | 0.22 |
| DGSAM | 54.5±0.6 | 45.3±0.7 | 59.4±0.4 | 42.3±1.0 | 50.4 | 6.9 | 0.17 |
| DGSAM + Mixup | 54.7±0.9 | 45.2±0.4 | 59.5±0.4 | 42.5±0.8 | 50.5 | 6.9 | 0.17 |
| DGSAM + CORAL | 55.8±0.5 | 45.4±0.8 | 59.2±0.2 | 42.7±1.1 | 50.8 | 6.9 | 0.19 |
| DGSAM + SWAD | 55.6±1.2 | 45.9±0.5 | 59.6±0.5 | 43.1±0.9 | 51.1 | 6.8 | 0.17 |
| DGSAM + ERM++ | 56.2±0.9 | 49.3±1.3 | 59.8±0.5 | 43.2±0.7 | 52.1 | 6.4 | 0.25 |

Table 13: The performance of DGSAM with 18 baseline algorithms on DomainNet

| Algorithm | C | I | P | Q | R | S | Avg | SD | (s/iter) |
|---|---|---|---|---|---|---|---|---|---|
| VREx[†] (Krueger et al., 2021) | 47.3 ±3.5 | 16.0 ±1.5 | 35.8 ±4.6 | 10.9 ±0.3 | 49.6 ±4.9 | 42.0 ±3.0 | 33.6 | 15.0 | 0.18 |
| ARM[†] (Zhang et al., 2021) | 49.7 ±0.3 | 16.3 ±0.5 | 40.9 ±1.1 | 9.4 ±0.1 | 53.4 ±0.4 | 43.5 ±0.4 | 35.5 | 16.7 | 0.18 |
| RSC[†] (Huang et al., 2020) | 55.0 ±1.2 | 18.3 ±0.5 | 44.4 ±0.6 | 12.2 ±0.2 | 55.7 ±0.7 | 47.8 ±0.9 | 38.9 | 17.3 | 0.20 |
| SagNet[†] (Nam et al., 2021) | 57.7 ±0.3 | 19.0 ±0.2 | 45.3 ±0.3 | 12.7 ±0.5 | 58.1 ±0.5 | 48.8 ±0.2 | 40.3 | 17.9 | 0.53 |
| MTL[†] (Blanchard et al., 2021) | 57.9 ±0.5 | 18.5 ±0.4 | 46.0 ±0.1 | 12.5 ±0.1 | 59.5 ±0.3 | 49.2 ±0.1 | 40.6 | 18.4 | 0.20 |
| ERM[†] | 58.1 ±0.3 | 18.8 ±0.3 | 46.7 ±0.3 | 12.2 ±0.4 | 59.6 ±0.1 | 49.8 ±0.4 | 40.9 | 18.6 | 0.18 |
| CORAL[†] (Sun & Saenko, 2016) | 59.2 ±0.1 | 19.7 ±0.2 | 46.6 ±0.3 | 13.4 ±0.4 | 59.8 ±0.2 | 50.1 ±0.6 | 41.5 | 18.3 | 0.20 |
| GENIE (Cho et al., 2025) | 62.5 ±0.5 | 21.3 ±0.4 | 50.0 ±0.4 | 14.0 ±0.4 | 64.0 ±0.7 | 52.6 ±0.8 | 44.1 | 19.4 | 0.14 |
| GGA (Ballas & Diou, 2025) | 63.7 ±0.2 | 21.3 ±0.3 | 50.4 ±0.1 | 14.1 ±0.4 | 63.8 ±0.2 | 53.5 ±0.3 | 44.4 | 19.7 | 0.75 |
| GGA-L (Ballas & Diou, 2025) | 63.2 ±0.2 | 21.0 ±0.3 | 49.5 ±0.1 | 13.8 ±0.2 | 64.1 ±0.4 | 53.6 ±0.3 | 44.5 | 19.7 | 0.50 |
| SWAD (Cha et al., 2021) | 66.0 ±0.1 | 22.4 ±0.3 | 53.5 ±0.1 | 16.1 ±0.2 | 65.8 ±0.4 | 55.5 ±0.3 | 46.5 | 19.9 | 0.18 |
| GAM[‡] (Zhang et al., 2023b) | 63.0 ±0.5 | 20.2 ±0.2 | 50.3 ±0.1 | 13.2 ±0.3 | 64.5 ±0.2 | 51.6 ±0.5 | 43.8 | 20.0 | 0.71 |
| Lookbehind-SAM (Mordido et al., 2024) | 64.3 ±0.3 | 20.8 ±0.1 | 50.4 ±0.1 | 15.0 ±0.4 | 63.1 ±0.3 | 51.4 ±0.3 | 44.1 | 19.4 | 0.71 |
| SAM[†] (Foret et al., 2021) | 64.5 ±0.3 | 20.7 ±0.2 | 50.2 ±0.1 | 15.1 ±0.3 | 62.6 ±0.2 | 52.7 ±0.3 | 44.3 | 19.4 | 0.34 |
| FAD (Zhang et al., 2023a) | 64.1 ±0.3 | 21.9 ±0.2 | 50.6 ±0.3 | 14.2 ±0.4 | 63.6 ±0.1 | 52.2 ±0.2 | 44.4 | 19.5 | 0.56 |
| GSAM[†] (Zhuang et al., 2022) | 64.2 ±0.3 | 20.8 ±0.2 | 50.9 ±0.0 | 14.4 ±0.8 | 63.5 ±0.2 | 53.9 ±0.2 | 44.6 | 19.8 | 0.36 |
| SAGM (Wang et al., 2023) | 64.9 ±0.2 | 21.1 ±0.3 | 51.5 ±0.2 | 14.8 ±0.2 | 64.1 ±0.2 | 53.6 ±0.2 | 45.0 | 19.8 | 0.34 |
| DISAM (Zhang et al., 2024) | 65.9 ±0.2 | 20.7 ±0.2 | 51.7 ±0.3 | 16.6 ±0.3 | 62.8 ±0.5 | 54.8 ±0.4 | 45.4 | 19.5 | 0.53 |
| DGSAM | 63.6 ±0.4 | 22.2 ±0.1 | 51.9 ±0.3 | 15.8 ±0.2 | 64.7 ±0.3 | 54.7 ±0.4 | 45.5 | 19.4 | 0.26 |
| DGSAM + CORAL | 64.3 ±0.2 | 22.5 ±0.2 | 54.2 ±0.3 | 16.2 ±0.2 | 64.9 ±0.1 | 55.2 ±0.2 | 46.2 | 19.5 | 0.28 |
| DGSAM + SWAD | 67.2 ±0.2 | 23.2 ±0.3 | 53.4 ±0.3 | 17.3 ±0.4 | 65.4 ±0.2 | 55.8 ±0.3 | 47.1 | 19.6 | 0.26 |
| DGSAM + Mixup | 67.4 ±0.3 | 25.4 ±0.1 | 54.8 ±0.2 | 17.6 ±0.3 | 67.5 ±0.4 | 57.3 ±0.3 | 48.3 | 19.7 | 0.26 |
| DGSAM + ERM++ | 71.3 ±0.3 | 26.9 ±0.2 | 58.6 ±0.2 | 17.9 ±0.5 | 70.5 ±0.2 | 60.8 ±0.5 | 51.0 | 20.9 | 0.43 |

## G    BASELINE REFERENCES

Table 1 compares our proposed method with several baseline algorithms for domain generalization. For a fair and consistent comparison, we report the performance metrics as presented in prior works.

Most results are sourced directly from the original papers introducing each algorithm. For certain baselines, results are quoted from recent state-of-the-art papers to ensure the experimental settings are as consistent as possible. Specifically, results marked with † are sourced from SAGM (Wang et al., 2023), and the result for GAM (‡) is from FAD (Zhang et al., 2023a).

The references for each baseline algorithm and combined methodology are as follows:

- ARM (Zhang et al., 2021)
- VREx (Krueger et al., 2021)
- RSC (Huang et al., 2020)
- MTL (Blanchard et al., 2021)
- SagNet (Nam et al., 2021)
- CORAL (Sun & Saenko, 2016)
- GGA & GGA-L (Ballas & Diou, 2025)
- GENIE (Cho et al., 2025)
- SWAD (Cha et al., 2021)
- GAM (Zhang et al., 2023b)
- SAM (Foret et al., 2021)
- Lookbehind-SAM (Mordido et al., 2024)
- GSAM (Zhuang et al., 2022)
- FAD (Zhang et al., 2023a)
- DISAM (Zhang et al., 2024)
- SAGM (Wang et al., 2023)
- SFT (Li et al., 2025)
- MixUp (Lopez-Paz et al., 2018)
- ERM++ (Teterwak et al., 2025)

## H    RELATED WORKS AND DISCUSSION

In this section, we complement the discussion in Section 2.2 by providing a more detailed categorization of SAM variants that have been applied to domain generalization. Our goal is to clarify how existing approaches interpret and optimize flatness in the multi-domain setting, and how this differs from the per-domain sharpness perspective underlying DGSAM.

**Domain-Agnostic Sharpness Minimization.**    This line of work adapts SAM or its extensions to DG by directly optimizing the aggregated sharpness. These algorithms do not utilize per-domain information and simply focus on reducing the sharpness of the aggregated loss, such as zero-th order sharpness or first-order sharpness.

For example, SAM and GAM, which were not originally designed for DG but are commonly used as baselines, reduce the zero-th order and first-order sharpness of the aggregated loss, respectively. FAM further aims to simultaneously reduce both zero-th order and first-order sharpness. On the other hand, GSAM, SAGM, and ISAM (Dong et al., 2024) are variants of SAM that reduce aggregated sharpness by mitigating gradient conflicts between the aggregated loss gradient and the surrogate gap, thereby achieving better reduction of aggregated sharpness. UDIM (Shin et al., 2024) introduces perturbations in both parameter space and data space for domain generalization. It reduces the loss landscape inconsistency between source domains and unknown domains, where unknown domains are emulated by perturbing instances from the source domain dataset. Although UDIM explores data

space perturbations, it does not utilize domain labels and ultimately optimizes for the consistency of aggregated loss landscapes.

**Domain-Aware Sharpness Minimization.** Another line of work explicitly incorporates domain labels into the sharpness optimization process, yet differs from our per-domain sharpness minimization approach.

DISAM (Zhang et al., 2024) introduces a domain loss variance regularization to achieve elastic gradient calibration: domains with higher losses receive weaker perturbations, while domains with lower losses receive stronger perturbations. This balancing mechanism promotes consistent convergence across domains, but the optimization still targets aggregated sharpness. Self-Feedback Training (SFT) (Li et al., 2025) seeks consistent flat minima across domains by iteratively measuring and refining loss landscape inconsistency. While it implicitly encourages per-domain flatness through consistency, it lacks a formal per-domain sharpness minimization formulation.

**Distinction and Novelty of DGSAM.** These two lines of work demonstrate that better control of aggregated sharpness and mitigation of domain inconsistency can improve DG performance. However, they still operate within the same objective: they ultimately seek to flatten the loss landscape of the aggregated source risk, sometimes with regularizers that indirectly promote consistency across domains.

By contrast, DGSAM starts from a DG-specific worst-case risk formulation and first asks a different question: "Is aggregated sharpness an appropriate surrogate for the average worst-case domain risk?" Our theoretical analysis shows that aggregated sharpness can be small even when some domains remain sharp, which gives rise to the fake flat minima phenomenon. We then prove that the average per-domain sharpness does provide a valid surrogate for the average worst-case domain risk.

This analysis yields an explicit per-domain sharpness objective whose minimizer is provably aligned with the DG goal, and DGSAM is designed as an algorithm that directly optimizes this objective while keeping the computational overhead practical. From a theoretical perspective, this provides a new way to think about sharpness in DG. Prior SAM-based DG approaches typically follow the original SAM line of analysis and study PAC-Bayes style bounds or regularization effects based on aggregated sharpness. In contrast, our work offers a new perspective on sharpness in DG by introducing a per-domain sharpness minimization framework that directly targets robustness to worst-case domains. We view this shift in objective as the main novelty of DGSAM and as a foundation for future sharpness-based methods in domain generalization.

