# OpenReview forum: "Global Sharpness-Aware Minimization Is Suboptimal in Domain Generalization: Towards Individual Sharpness-Aware Minimization"
_ICLR.cc/2026/Conference — Submitted to ICLR 2026_

### Official Review · Reviewer_E45h · 2025-10-15

**Soundness:** 3
**Presentation:** 3
**Contribution:** 3
**Rating:** 8
**Confidence:** 5

**Summary:**

This paper presents an idea of using individualized SAM to mitigate the domain generalization problem. Authors call the previous SAM approach to be "global", which I am uncomfortable because SAM is originally an idea of local flat minima (having said that I understand the authors claim and I concur that "aggregated" loss can be deceiving flat minima). They make an individualaity to represent a certain domain. Then, they present a method named DGSAM to control the individual sharpness. Experiments support the claim well.

**Strengths:**

1. Very nice problem discovery of the aggregated loss from the SAM perspective.
2. Good principled approach to define the domain generalization and its relation toward the individual sharpness
3. Necessary proofs are all provided. The defined individual sharpness will reduce the domain generalization errors. Stationary aspect, and its derived optimization approach path.

Classic problem definition and solving it.

**Weaknesses:**

1.
I am very uncomfortable with the terminology that they defined or used.
As I mentioned earlier in the summary, all SAM approaches assume the parameters will be favored if they are located at the flat minima. However, this flat minima in the parameter space is always epsilon small, so SAM is always localized approach to a certain extent controlled by the epsilon. If authors agree with this aspect, then they would agree calling "global sharpness" is in its contradiction. What they are really pointing out is "the aggregated flat-minima loss surface over the parameter space". I would rather use "aggregation" instead of using a word "global".

2.
"Decreased-overhead" is your methodology name. I partially agree that the computational requirement could be reduced in the line of SAM researches. Having said that, I don't think that the overhead decrement would be your key contribution throughout the paper. Your key contribution is treating the "domain-specific" parameter space perturbation before "domain-aggregation" (yes. I don't like your wording 'individual' either), whereas the previous approaches have been "domain-aggregated" parameter space perturbation. Does this reversed process reduce the overhead? Could be. Is it the main-theme? No.

**Questions:**

I don't have much question on this paper. I think that I understand enough to see the merit of this paper. I would like to get answers from my weakness section.

**Details Of Ethics Concerns:**

pure methodology. no ethics needed

---

> ### Author Response · Authors · 2025-11-20
>
> We thank the reviewer for the positive assessment and the constructive feedback. We appreciate the reviewer’s careful comments on our work.
>
> Below, we address the concerns raised in the Weaknesses section.
>
>
> > **W1. Terminology regarding “global sharpness”**
>
>
> Thank you for this thoughtful comment. We fully agree that this terminology may cause confusion, and we appreciate the reviewer’s suggestion that expressions such as “aggregated loss sharpness” might more clearly convey the intended distinction. Building on this, we see two reasonable options for revision:
>
> 1. **Option 1 (clarification without renaming).**
>    We keep the terminology “global sharpness,” but explicitly define it early in the paper as “sharpness of the loss aggregated across domains” and emphasize this meaning wherever the term appears. We can, for example, add a short clarifying sentence or a small explanatory table to clearly distinguish “global sharpness” from “average of domain specific sharpness,” so that readers are not misled into interpreting “global” as a spatially non-local notion.
>
> 2. **Option 2 (full renaming).**
>    We adopt the reviewer’s suggestion more directly and replace “global sharpness” with a term such as “aggregated sharpness” throughout the paper, including the title if appropriate. According to the ICLR 2026 policy, such changes are permitted as long as they are clearly communicated to the reviewers and the area chair, so we can revise the terminology in a coordinated manner.
>
> If the reviewer feels that Option 1 would still be insufficient and that a more substantial renaming is necessary, we are happy to follow Option 2 in agreement with the area chair and the other reviewers.
>
>
> > **W2. Naming related to “decreased-overhead”**
>
>
>
> Thank you for pointing this out. We agree that emphasizing “decreased overhead” in the name may draw attention away from the main idea, which is the domain-wise perturbation prior to aggregation. In line with this feedback, we have renamed the method to **Domain-wise Gradual SAM** in the revised manuscript. We believe this better reflects the core contribution.

---

> > ### Comment · Reviewer_E45h · 2025-11-20
> >
> > If you follow option 1, why don't you use "domain-generalized sharpness"? "global" has a very specific meaning in the optimization world, particularly when it is compared to the locality, i.e. SAM.
> >
> > Anyhow, you need to respect the terminology usage in the line of research

---

> ### Author Response · Authors · 2025-11-20
>
> Thank you for your quick response. Understood. We will revise the terminology throughout the manuscript by replacing “global sharpness” with “aggregated sharpness” and “individual sharpness” with “per-domain sharpness” to ensure clarity and consistency. Since this also affects the title, we will communicate the title change through the global response so that the other reviewers and the area chair are fully informed.
>
> Thank you again for your careful and constructive comments.

---

### Official Review · Reviewer_QcVt · 2025-10-28

**Soundness:** 3
**Presentation:** 3
**Contribution:** 3
**Rating:** 4
**Confidence:** 4

**Summary:**

This work examines how Sharpness-Aware Minimization (SAM) should be applied in the context of domain generalization (DG), and it challenges the conventional “global” use of SAM on aggregated training loss. The authors point out that simply finding a flat minimum for the average loss over all source domains can be misleading: it may produce a “fake flat” solution that appears robust overall but still has sharp (high-curvature) loss landscapes on individual domains, leaving the model vulnerable to domain-specific shifts.

To address this, they analyze a worst-case (adversarial) risk formulation for DG and show theoretically that minimizing individual-domain sharpness provides an upper bound on this worst-case risk, whereas minimizing global sharpness does not. In other words, each source domain’s loss landscape needs to be flat for true robustness, not just the combined loss.

Building on this insight, the paper proposes DGSAM (Decreased-overhead Gradual SAM), which explicitly targets sharpness on a per-domain basis while keeping computational cost manageable.

**Strengths:**

Empirical results on five standard DG benchmarks show that DGSAM achieves better overall accuracy and significantly lower performance variance across domains compared to both standard training and globally-applied SAM.

Models trained with DGSAM are consistently more robust to unseen target domains, indicating that the individual sharpness objective indeed translates to improved domain generalization.

Moreover, DGSAM is shown to be computationally efficient – it incurs less overhead than traditional SAM (which doubles the compute) – and scales to large architectures like Vision Transformers.

The paper’s contribution is notable in reframing SAM for multi-domain settings and providing both a theoretical justification and a practical algorithm that improves robustness.

**Weaknesses:**

DGSAM adds algorithmic complexity by requiring domain-specific updates (which could scale in cost with the number of domains), but the authors claim that they mitigate this with their gradual update scheme. Would this also fit with the case when the number of domains get really high (about 100 ~ )?

It would be beneficial to discuss more about related SAM works.
- https://arxiv.org/abs/2410.14802 : Discussion about data-responsive regularization, and why still per-domain sharpness is required?
- https://arxiv.org/abs/2403.07329 : How DGSAM differs from UDIM, when UDIM tries to generalize toward unseen domain?

I also want authors to measure "the zeroth-order sharpness result at converged minima" not only compared to original SAM, and other algorithms. To more precisely compare the impact of sharpness.

I think that this method is somewhat incremental from existing methods, because there were heavy-amount of SAM variants for domain generalization. But, if questions above are treated well, i will change my score.

**Questions:**

DIscussed in weaknesses section.

---

> ### Author Response · Authors · 2025-11-20
>
> We sincerely thank the reviewer for the constructive comments and helpful suggestions. Below, we respond to the point raised in the Weaknesses section.
>
> >**W1. Scalability when the number of domains is large**
>
> We appreciate the reviewer’s important question regarding the scalability of DGSAM when the number of source domains becomes very large. A naive implementation that visits all $S$ domains at every iteration can indeed be inefficient in such cases. To handle this setting, Appendix C.4 in original manuscript introduces a simple extension based on domain subsampling.
>
> Instead of performing sequential ascent over all S domains, we fix the number of ascent steps to a small constant $k$ (with $k \ll S$, for example $k = 5$) and randomly sample $k$ domains at each iteration. The algorithm in the main text corresponds to the special case $k = S$, and the subsampling variant is intended for scenarios with many domains.
>
> Table 5 in our manuscript shows that this variant of DGSAM maintains strong performance on datasets with several tens of domains, such as PovertyMap (23 domains) and GlobalWheat (47 domains), while keeping the computational cost controlled. This supports that DGSAM, together with domain subsampling, can be used in practice when the number of domains is much larger.
>
>
> >**W2. Discussion of suggested references**
>
> We thank the reviewer for pointing us to these relevant studies. Li et al. analyze SAM’s implicit regularization on a single distribution and show that SAM becomes more data-responsive under low-SNR or noisy settings. UDIM similarly models domain shift by introducing synthetic perturbations to a single dataset. These works analyze SAM through the lens of global sharpness.
>
> In contrast, our work identifies a more structural issue specific to domain generalization: global sharpness computed on the aggregated loss can mask sharp regions that persist on individual domains. We show theoretically that this mismatch creates the fake flat minima phenomenon and that per-domain sharpness aligns with the average worst-case domain risk.
>
> This leads to a novel perspective compared to prior SAM-based approaches: robustness in DG requires controlling sharpness at the level of individual domains rather than relying on aggregated sharpness used in prior work. We will incorporate a discussion of these connections and the suggested papers in the revised manuscript
>
>
>
>
>
> >**W3. Comparison of sharpness measures across algorithms**
>
> We thank for your valuable suggestion. In the revised version, we compare sharpness at the converged minima across ERM, SAM, DISAM, SAGM, and our method, using zeroth-order sharpness, the maximum eigenvalue, and the Hessian trace. The results in Table 3 show that DGSAM consistently achieves lower per-domain sharpness across all metrics, which further supports the motivation behind our formulation.

---

> ### Author Response · Authors · 2025-11-21
>
> > **W4. Novelty and conceptual contribution**
>
> Thank you very much for raising this point. As the reviewer mentioned, many SAM variants, such as SAGM, ISAM, DISAM, and UDIM, improve the optimization behavior of SAM in DG settings by addressing specific issues within the aggregated SAM objective.
>
> Our work is conceptually different. Rather than proposing another modification of the SAM update rule, we revisit the objective of sharpness-aware training in DG and ask a more fundamental question:
>
> **"Is aggregated sharpness an appropriate surrogate for the worst-case domain risk?"**
>
> Our theoretical analysis shows that the answer is no. Sharpness computed from the aggregated loss can appear small even when some domains remain sharp, creating a failure mode we term fake flat minima. This reveals a structural limitation of using aggregated sharpness in DG and implies that robustness requires controlling sharpness at the level of each domain, not only on the aggregated loss.
>
> Motivated by this finding, we derive a per-domain sharpness objective that aligns with the average worst-case DG risk and propose DGSAM as an algorithm that directly optimizes this formulation. In this sense, our contribution is not an incremental refinement of SAM’s mechanics but a shift in perspective that reframes how sharpness should be modeled in multi-domain settings. We believe this framework has the potential to inspire a broader family of algorithms built upon per-domain sharpness, rather than representing a single isolated SAM variant.

---

> > ### Comment · Reviewer_QcVt · 2025-11-27
> > **Response to the Authors**
> >
> > I appreciate authors to examine several parts to resolve my concerns.
> > I reviewed the revised manuscripts and also your response, and most of my concerns were resolved.
> > So i am increasing my score to 6.

---

### Official Review · Reviewer_JpK6 · 2025-11-06

**Soundness:** 3
**Presentation:** 3
**Contribution:** 2
**Rating:** 4
**Confidence:** 3

**Summary:**

The paper argues that the common practice of applying SAm to the global loss across source domains (in the context of DG) is suboptimal, i.e., it can lead models toward fake flat minima that remain sharp on individual domains. The authors introduce an average worst-case risk formulation and prove that individual sharpness per-domain yields a valid upper bound to this risk, whereas global sharpness does not. In response, the authors propose DGSAM, a gradual, domain-wise perturbation method that controls individual sharpness with lower cost than SAM and improves both average acc. and cross-domain variance.

**Strengths:**

Thank you for your submission, I enjoyed reading the paper and the fresh ideas on generalization from the loss landscape perspective. Below I have listed some aspects of the paper I've appreciated.

- The paper points a conceptual flaw in how SAM has been ported to DG: "optimizing flatness of all the domains combined is not the same as ensuring generalization for each domains shift". From that perspective, the authors' idea that "individual domain sharpness is the right surrogate for DG" is sound and convincing.
- The proposed algorithm DGSAM aligns well with the problem formulation, and its effectiveness is supported through multiple empirical gains over a number of standard DG benchmarks. Also, the gradient re-use is notable, its efficiency was shown empirically (Sec. 5.4).
- From my understanding, this paper is an extension of previous works in two key ways: (1) The authors suggest that minimizing the average of per‑domain sharpness provides a valid upper bound, while minimizing the global (aggregated) sharpness does not guarantee flat minima, and thus in improving generaliztion. Prior works do not explicitly mention the issue of fake flat minima. (2) It proposes a sequential, domain‑wise perturbation scheme that reuses gradients, so each domain’s loss surface is explicitly flattened.
- Overall, the paper was easy to read and the message was clear. The theoretical analysis was also sound (Sec.3) and in accordance with the previous literature.

**Weaknesses:**

- Positioning: While the paper aims to address an important, yet often overlooked issue in applying SAM to DG (or approaching generalization from the loss landscape perspective), it is still close to previous works (SAGM, ISAM, DISAM) that are aware of the issues in the naive application of SAM. Each of these works modifies the SAM objective to address specific deficiencies (e.g., inaccurate sharpness measures and gradient conflicts -- ISAM, inconsistent convergence across domains -- DISAM). Although they are cited and included among baselines, the paper would largely benefit from a sharper positioning.
    - For instance, the idea that 'global sharpness' and 'per-domain sharpness' may not align was previously observed in Le et al. (2024). Although the paper was cited in line 484, we believe that their observation should be further noted as it aligns with the core idea of the paper. Similarly, the global vs. local sharpness/flatness idea was also studied in the federated learning literature [1].

- Cost Measure: A minor one, but in the paper, per-iteration gradient counts are reported (Sec 5.4), but end-to-end wall-clock, FLOPs, and peak memory comparisons are missing. Could the authors provide this? Again, this is a minor suggestion.

- Statistical Stability: Also a minor one, but in the camera-ready version, we suggest the authors to provide the average performance and standard error across more than 3 runs.

- Sharpness: In Sec 5.3 (and Tab. 3), the zeroth-order sharpness is measured to show that DGSAM can effectively reach flat minima. To my understanding, the zeroth-order sharpness refers to the maximal loss within the perturbed neighborhood. While they are a useful objective, they are still limited proxies that have distinct limitations [2,3]. In the generalization literature, different metrics are also commonly used (e.g., the largest eigenvalue of the Hessian), owing to their theoretical implications.
    - In response, I believe that the authors should supplement their analysis with additional diagnostics. e.g., reporting the maximum Hessian eigenvalue or trace per domain (or at least proxies such as top‑eigenvalue estimates) would provide stronger evidence that the method genuinely finds flatter minima.

***
### Reference

[1] Caldarola et al., Beyond Local Sharpness: Communication-Efficient Global Sharpness-aware Minimization for Federated Learning, CVPR, 2025.

[2] Zhuang et al., Surrogate gap minimization improves sharpness-aware training, ICLR, 2022.

[3] Bian et al., Make Continual Learning Stronger via C-Flat, NeurIPS, 2024.

**Questions:**

- Expansion: One small question is whether the method can also be applied to single-source settings (Single-source Domain Generalization). My guess is that it wouldn't work (simply out of scope!) and would collapse to SAM, unless there are simulated (commonly augmented) domains. I acknowledge that the paper focuses on multi-domain settings, and this question is purely out of curiosity.

- Ablation Study: I'm interested in several components of the method and their effect on the performance gains. For instance, (1) what happens if the domain order is fixed, instead of being random (Line 3 in Algorithm 1)? (2) re-using vs. not re-using the ascent gradients.

Please refer to the Weaknesses section for the questions. I'm mostly interested in the Sharpness measure and the Ablation study.

---

> ### Author Response · Authors · 2025-11-20
>
> We sincerely appreciate the reviewer’s detailed comments and helpful suggestions. We address the concerns and questions raised in the review in the responses below.
>
> - **Response to Weaknesses (W)**
>
> > **W1. Positioning with respect to prior SAM variants**
>
> We are grateful for the opportunity to clarify the positioning of our contribution. We emphasize that our contribution is not a simple or incremental modification of SAM designed for DG tasks. Instead, we revisit the objective of sharpness-aware training in domain generalization and ask a more fundamental question:
> “Is global sharpness a valid surrogate for worst-case domain risk?”
>
> Our theoretical analysis shows global sharpness does not serve as a valid surrogate for the worst-case DG risk. Aggregated losses can appear flat even though some domains remain sharp, and this discrepancy leads to a failure mode that we refer to as fake flat minima. This reveals a structural limitation of using global sharpness in DG and shows that sharpness must be controlled on each domain individually.
>
> Motivated by this analysis, we derive a per-domain sharpness objective and develop an algorithm that directly optimizes this theoretically grounded formulation. This viewpoint is complementary to prior SAM-based approaches such as SAGM, ISAM, and DISAM. As the reviewer mentioned, these methods refine the optimization behavior of SAM for DG in various ways while continuing to operate on the aggregated SAM objective. In contrast, our work focuses on a different question at the objective level: whether aggregated sharpness itself is appropriate for domain generalization. This objective-level difference provides the basis for our new perspective on domain generalization.
>
> We appreciate the reviewer’s pointer to related observations in federated learning. These works study local–global curvature mismatch caused by client averaging, which is a different setting. Our analysis focuses on the opposite failure mode: the aggregated loss can appear flat even when individual domains remain sharp. Based on this, we formally characterize the fake flat minima phenomenon and derive a domain-wise sharpness objective that is suitable for worst-case DG risk.
>
> > **W2. Cost measures**
>
> We appreciate this helpful suggestion. The original manuscript already includes an analysis of memory usage, covering both mean and peak consumption in Appendix D.2 (Table 6). In the revised version, we additionally include a brief comparison of computational cost, including FLOPs, to provide a clearer overview of the overall cost profile.
>
>
> > **W3. Statistical stability**
>
> Thank you for pointing this out. Our experiments follow the common protocol in the DG literature, where methods such as SWAD, SAGM, and DISAM also report results based on three runs. We will consider increasing the number of runs in the camera-ready version and, if feasible, provide the mean and standard error to offer a more reliable estimate of statistical variability.
>
>
>
> > **W4. Additional sharpness metrics**
>
> Thank you for this helpful comment. As shown in Figure 3 of the original manuscript, we already analyze the Hessian spectrum and observe that DGSAM consistently yields lower largest eigenvalues than SAM across individual domains, which aligns with the motivation of our method.
>
> In the revised manuscript, we extend this analysis by comparing sharpness at the converged minima across ERM, SAM, DISAM, SAGM, and DGSAM using three measures: zeroth-order sharpness, the maximum eigenvalue, and the Hessian trace. The results, summarized in Table 3, show that DGSAM consistently achieves lower individual domain sharpness across all metrics, providing further evidence that our method reaches flatter minima for each domain.

---

> ### Author Response · Authors · 2025-11-20
>
> - **Response to Questions (Q)**
>
>
>
> > **Q1. Expansion**
>
> Thank you for raising this interesting question. DGSAM is designed for multi-domain settings, where the distinction between the aggregated loss and the per-domain sharpness becomes meaningful. When only a single source domain is available, this distinction disappears, and the method collapses to standard SAM, as the reviewer anticipated. Creating pseudo-domains through augmentations is a possible extension, but such scenarios fall outside the scope of the current work.
>
>
> > **Q2. Ablation study**
>
> Thank you for your constructive suggestion. We have conducted the suggested ablation studies on the PACS and TerraIncognita datasets and included the results in Appendix C.5 of the revised manuscript.
>
> 1. **Random vs. Fixed Domain Order**:
> We compared our default random permutation strategy with a "Fixed Order" variant where the domain sequence remains constant throughout training. Our experiments reveal that fixing the domain order leads to a consistent degradation in average accuracy and a marked increase in performance variance across domains. These results suggest that the randomness in domain ordering acts as a crucial regularizer, preventing the optimization from biasing towards a specific domain and ensuring robust flatness across all domains.
>
> 2. **Re-using vs. Not Re-using Gradients**:
> We investigated the trade-off between our gradient reuse strategy and a variant that performs a fresh gradient computation for the final update ("Not re-using"). We found that while recalculating gradients yields marginal performance gains, it incurs a substantial increase in computational overhead and training time. Consequently, we demonstrate that the gradient reuse strategy provides a much better balance between predictive performance and computational efficiency for scalable domain generalization.

---

> ### Comment · Reviewer_JpK6 · 2025-11-25
>
> I appreciate the authors’ rebuttal, as well as the additional content added in the revision that helps clarify several concerns I had. That being said, I feel that the positioning relative to prior SAM-based DG methods is still somewhat under-emphasized, and I believe the paper could convey this distinction more clearly. On a different note, the revised sharpness evaluations are helpful, and while I am not yet fully convinced that the empirical evidence around “fake flat minima’’ and per-domain flatness, I appreciate the authors’ efforts to expand this analysis. For these reasons, primarily the remaining questions about distinctiveness relative to existing SAM X DG works, I am inclined to keep my overall score unchanged. However, I will make a final decision after reading all reviewers’ responses and evaluating the paper holistically in the discussion phase.

---

> ### Author Response · Authors · 2025-11-27
>
> >**Positioning relative to prior SAM-based DG methods:**
>
> Thank you again for your careful follow-up. We have carefully strengthened the positioning to directly address this concern and clarified that while existing SAM based DG methods still rely on aggregated sharpness, DGSAM is **the first to theoretically justify and optimize per-domain sharpness for DG**. Importantly, our analysis does not merely refine how SAM is applied to DG; it revisits what sharpness should mean in DG and identifies per-domain sharpness as the correct quantity to control.
>
> More specifically, we have added a new section H in the appendix that categorizes SAM variants used for DG into two groups:
>
> (i) domain-agnostic sharpness minimization, which always operates on aggregated sharpness (SAM, GAM, FAM, GSAM, SAGM, ISAM, UDIM), and
>
> (ii) domain-aware methods that incorporate domain labels but still optimize aggregated sharpness or lack a formal per-domain objective (DISAM, SFT).
>
> By contrast, DGSAM starts from a DG-specific worst-case risk formulation and first asks a different question: "Is aggregated sharpness an appropriate surrogate for the average worst-case domain risk?" Our theoretical analysis shows that aggregated sharpness can be small even when some domains remain sharp, which gives rise to the fake flat minima phenomenon. We then prove that the average per-domain sharpness does provide a valid surrogate for the average worst-case domain risk.
>
> This analysis yields an explicit per-domain sharpness objective whose minimizer is provably aligned with the DG goal, and DGSAM is designed as an algorithm that directly optimizes this objective while keeping the computational overhead practical. From a theoretical perspective, this provides a new way to think about sharpness in DG. Prior SAM-based DG approaches typically follow the original SAM line of analysis and study PAC-Bayes style bounds or regularization effects based on aggregated sharpness. In contrast, our work offers a new perspective on sharpness in DG by introducing a per-domain sharpness minimization framework that directly targets robustness to worst-case domains. We view this shift in objective as the main novelty of DGSAM and as a foundation for future sharpness-based methods in domain generalization.
>
>
> We have highlighted this contrast more clearly in Section 2.2 and in the new appendix section H to better convey how our formulation differs from existing SAM-based DG approaches.
>
>
> >**Empirical Evidence of Fake Flat Minima**
>
> Thank you very much for your thoughtful follow-up on this point. We understand the concern and would like to clarify how the current evidence supports our interpretation.
>
>
> In Figure 9, we visualize the loss landscapes for SAM and DGSAM across different domains on PACS. While both methods achieve flat total loss landscapes, SAM exhibits sharp regions in individual domains (particularly Art and Sketch), whereas DGSAM achieves consistent flatness across all domains.
>
> Table 3 further highlights a key pattern. Although SAGM attains a lower aggregated Hessian trace than DGSAM, DGSAM achieves lower per-domain Hessian trace, both in terms of the mean and the standard deviation across domains. **This is precisely the fake flat minima phenomenon we describe, where the aggregated loss surface appears flat while some domains remain sharp.** Importantly, DGSAM also outperforms SAGM on DomainNet in Table 1, which suggests that per-domain sharpness is more closely tied to domain generalization performance than aggregated sharpness alone.

---

### Author Response · Authors · 2025-11-20
**Global Response: Clarification on Terminology Update for the Revised manuscript ("Aggregated Sharpness" & "Per-domain Sharpness")**

Dear reviewers and area chair,

Thank you for your constructive feedback. After discussing the terminology issue raised by Reviewer E45H, we recognize that the terms may introduce unnecessary confusion. To improve clarity, we will update the terminology throughout the manuscript as follows:

- *global sharpness* → **aggregated sharpness**
- *individual sharpness* → **per-domain sharpness**

These updates also require modifying the paper title. In accordance with the ICLR 2026 policy, such changes should be communicated to all reviewers and the area chair, so we are sharing this revision plan here.

If there are any potential issues we may have overlooked regarding this terminology update, we would appreciate your guidance.

Thank you again for your time and for your careful, high-quality reviews.

---

### Author Response · Authors · 2025-12-02

We sincerely appreciate the reviewers’ constructive feedback, which has significantly improved both the rigor and clarity of our manuscript. Below, we summarize the major revisions included in the updated version.

>**1. Expanded sharpness evaluation** *(response to reviewers JpK6, QcVt)*

We broadened the sharpness analysis by incorporating:
- additional baselines: ERM, SAM, SAGM, DISAM
- multiple metrics: zeroth-order sharpness, Hessian trace, maximum eigenvalue

The results, presented in Table 3 of the updated manuscript, further demonstrate that DGSAM consistently achieves lower per-domain sharpness across all evaluated metrics.

>**2. Terminology & algorithm naming revision** (response to Reviewer E45h)

To avoid ambiguity and ensure consistency with established terminology, we renamed:
- global sharpness $\rightarrow$ aggregated sharpness
- individual shaprness $\rightarrow$ per-domain sharpness

We also updated the algorithm name to **Domain-wise Gradual SAM (DGSAM)** to more clearly reflect its key mechanism.


>**3. Positioning and novelty** *(response to JpK6, QcVt)*

We carefully strengthened the positioning to address this point directly. In particular, we now clarify that the core novelty of DGSAM does not merely lie in improving the SAM update rule, but in reformulating the sharpness objective itself for domain generalization in a way that fundamentally redefines what sharpness should mean in DG.

We added a dedicated new section in Appendix H (Related Works and Discussion) that categorizes existing SAM × DG methods into:
- Aggregated sharpness optimization (SAM, GAM, FAM, GSAM, SAGM, ISAM, UDIM)
- Domain-aware approaches that still operate on aggregated sharpness or without a formal per-domain objective (DISAM, SFT)

Our analysis challenges the common assumption that aggregated sharpness is the correct DG objective and theoretically establishes per-domain sharpness as the appropriate surrogate for worst-case DG risk. We believe this represents a shift in perspective, laying the foundation for future sharpness-based DG approaches.

>**4. Ablation studies** *(response to Reviewer JpK6)*

Appendix C.5 now includes ablations on:

1. Random vs. fixed domain ordering
2. Gradient reuse vs. recomputation

These results support our intended design trade-off: DGSAM retains most of the performance benefits while keeping the computational overhead practical.

>**5.computational cost and scailability** *(response to Reviewer JpK6, QcVt)*

Building on prior reports of faster training speed (iterations per second) and lower peak memory usage, we additionally include a FLOPs comparison in Appendix D.2, offering a more comprehensive cost profile.


Regarding scalability when the number of domains becomes large, experimental results with domain subsampling can be found in Appendix C.4, which indicate that DGSAM remains practical even in scenarios with numerous domains while maintaining strong performance.

-------------------------------------------------------------
----------------------------------------------------------


Dear Area Chair,

Thank you very much for taking the time to handle our submission.

In light of the recent OpenReview incident, we would like to provide you with a concise summary of the reviews of our paper and our responses. For the avoidance of doubt, we have not accessed, shared, or made use of any leaked identity information and have strictly adhered to the double-blind reviewing rules throughout the process. This note is solely intended to help you quickly reconstruct the discussion around the submission. We once again sincerely appreciate your time and effort in overseeing the review process despite the difficulties caused by this incident.

Below, we provide a summary of the reviewers’ evaluations and their responses during the rebuttal phase.


**Reviewer Rating Summary**

| Reviewer | Initial Rating (Conf.) | Rebuttal Outcome (Conf.) | Notes |
|---------|----------------------------|-----------------|----------------------|
| E45h | 8 (5) | 8 (5) | Fully satisfied after terminology & naming updates |
| QcVt | 4 (4) |  6 (4) | Concerns resolved after additional experiments & analysis |
| JpK6 | 4 (3) | Under reconsideration | Acknowledged several improvements; requested further strengthening of the novelty positioning before making a final decision, but the process was disrupted by the OpenReview incident |

---

### Meta-Review · Area_Chair_5MpF · 2026-01-06

**Summary:**

The submission examines the difference in encouraging flatness for two different domain generalisation objectives: one where the loss for each domain is aggregated linearly, and another where a different worst-case perturbation is added to each domain first, as is done in distributionally robust optimisation (DRO). It is shown formally that there are cases where one cannot find a point that is simultaneously a flat minima of both training objectives. The paper then explores the performance and behaviour of a Sharpness-Aware Minimisation (SAM) method for optimising the DRO-style objective.

The biggest concern raised by reviewers is the minimal novelty of investigating SAM for domain generalisation, and the lack of detail positioning the contributions of this paper relative to existing work. There were also concerns related to the empirical investigation into the mechanism for why the proposed method might work; it was suggested a more comprehensive investigation in to the sharpness of solutions produced by this method and other methods is warranted. One reviewer also identified several minor weak points related to the presentation of the work.

**Reviewer Concerns:**

Many of the minor concerns put forth by the reviewers have been adequately resolved. However, I do not think the central issue of it being unclear what new knowledge this work provides over existing papers has been addressed. In particular, the SFT work of Li et. al. (2025) seems to be exploiting precisely the same intuition as the proposed approach. The existing discussion does not clearly elucidate what this submission adds over that previous work, but suggests that this work has stronger theoretical contributions. The main justification for investigating the per-domain sharpness strategy is that it is more aligned with the DRO-style objective ($\mathcal{E}$) that the authors propose, but there is no theoretical justification provided for why we should care about this objective more than the usual DG objectives. Moreover, the experimental analysis still reports mean accuracy across domains as the metric of interest, which is more aligned with the aggregated objective.

Another important issue that none of the reviewers picked up on, and made a minor contribution to my decision, is the hyperparameter optimisation process used in the experimental component of this work. Looking at the submitted code, the authors have specified different search spaces for learning rate and weight decay for their method compared to all other methods in domain bed. No discussion is provided in the paper for why the proposed method should need a different hyperparameter search space for these quantities. This might seem like a non-issue, but the HPO method (20 iterations of random search) used by DomainBed is not very sophisticated, so providing a more constrained/informative search space for some hyperparameters can make a real difference to the resulting performance comparison. Given the quite small margins between the performance of the proposed approach and the performance of existing methods, this is something that should be addressed.

**Reviewer Scores:**

QcVt stated themselves that they would update their score to a 6. I do not believe the concern of JpK6 was sufficiently addressed to elicit an updated score.

---

### Decision · Program_Chairs · 2026-01-26

Reject